

# Effects of mediated social touch on affective experiences and trust

Stefanie M. Erk[1], Alexander Toet[1,2] and Jan B.F. Van Erp[1,3]

[1] TNO, Soesterberg, Netherlands
[2] Experimental Psychology, Helmholtz Institute, Utrecht University, Utrecht, Netherlands
[3] Human Media Interaction, University of Twente, Enschede, Netherlands

Corresponding author
Alexander Toet, lextoet@gmail.com

## ABSTRACT

This study investigated whether communication via mediated hand pressure during a remotely shared experience (watching an amusing video) can (1) enhance recovery from sadness, (2) enhance the affective quality of the experience, and (3) increase trust towards the communication partner. Thereto participants first watched a sad movie clip to elicit sadness, followed by a funny one to stimulate recovery from sadness. While watching the funny clip they signaled a hypothetical fellow participant every time they felt amused. In the experimental condition the participants responded by pressing a hand-held two-way mediated touch device (a Frebble), which also provided haptic feedback via simulated hand squeezes. In the control condition they responded by pressing a button and they received abstract visual feedback. Objective (heart rate, galvanic skin conductance, number and duration of joystick or Frebble presses) and subjective (questionnaires) data were collected to assess the emotional reactions of the participants. The subjective measurements confirmed that the sad movie successfully induced sadness while the funny movie indeed evoked more positive feelings. Although their ranking agreed with the subjective measurements, the physiological measurements confirmed this conclusion only for the funny movie. The results show that recovery from movie induced sadness, the affective experience of the amusing movie, and trust towards the communication partner did not differ between both experimental conditions. Hence, feedback via mediated hand touching did not enhance either of these factors compared to visual feedback. Further analysis of the data showed that participants scoring low on *Extraversion* (i.e., persons that are more introvert) or low on *Touch Receptivity* (i.e., persons who do not like to be touched by others) felt better understood by their communication partner when receiving mediated touch feedback instead of visual feedback, while the opposite was found for participants scoring high on these factors. The implications of these results for further research are discussed, and some suggestions for follow-up experiments are presented.

## INTRODUCTION

### Aim of this study

Touch provides a powerful means of evoking and modulating human emotion. Touching someone—as well as being touched by someone—can have a positive and calming effect. People who are touched more often by their partner also report better psychological well-being (*Debrot et al., 2013*; *Debrot et al., 2014*). From nursing care it is known that human touch can promote physical, emotional, social and spiritual comfort (*Chang, 2001*; *Whitcher & Fisher, 1979*; see also *Field, 2010*), for instance by effectively reducing worries (*Whitcher & Fisher, 1979*), anxiety and pain (*Anderson, 2001*).

In daily life the action of touching or holding hands is an important empathic experience for dyads that elicits a strong sense of togetherness, while signaling trust, understanding and social support (*Field, 2010*). Haptic devices that mediate the act of holding hands via the internet may therefore provide a means to foster a powerful sense of intimacy and connectedness between physically separated dyads (*Toet et al., 2013*; *Van Erp & Toet, 2015*). The present study was performed to investigate whether mediated touch can elicit some of the beneficial effects reported for direct touch and can therefore be used to improve the quality of interpersonal contact and communication at a distance. In particular, this study investigates whether sharing emotions via mediated social touch can (1) enhance recovery (i.e., the return to a more positive emotional state) after a negative (sad) experience, (2) enhance a positive (amusing) experience, and (3) increase trust towards the communication partner. At this stage the focus of our research is primarily on eliciting affective responses similar to those that occur while holding hands, and not so much on a realistic simulation of the feeling of a human hand (skin texture and temperature).

### The importance of interpersonal touch

Touch is the preferred nonverbal communication channel for conveying intimate emotions like love and sympathy (*App et al., 2011*). Interpersonal touch can promote physical, emotional, social and spiritual wellbeing (*Field, 2010*). The gentle touch from another person influences our readiness to empathize with and support that person (*Guéguen & Fischer-Lokou, 2003*). A handshake, an encouraging pat, a sensual caress, a nudge for attention, or a gentle brush can all convey a vitality and immediacy that is at times far more powerful than language (*Jones & Yarbrough, 1985*).

Geographically separated people in need of social support often long for each other's physical presence. Typical examples are family members who are separated by great distances, such as grandparents who live far from their grandchildren, a child who is sick in a hospital far from family members, a parent who is away on a business trip, people with parents in nursing homes, etc. The development and diffusion of internet-based technologies has enabled people who are globally separated to easily interact with each other. However, most of the currently available technologies do not include the tactile aspects of interpersonal communication, which are known to play a crucial role in establishing a sense of togetherness.

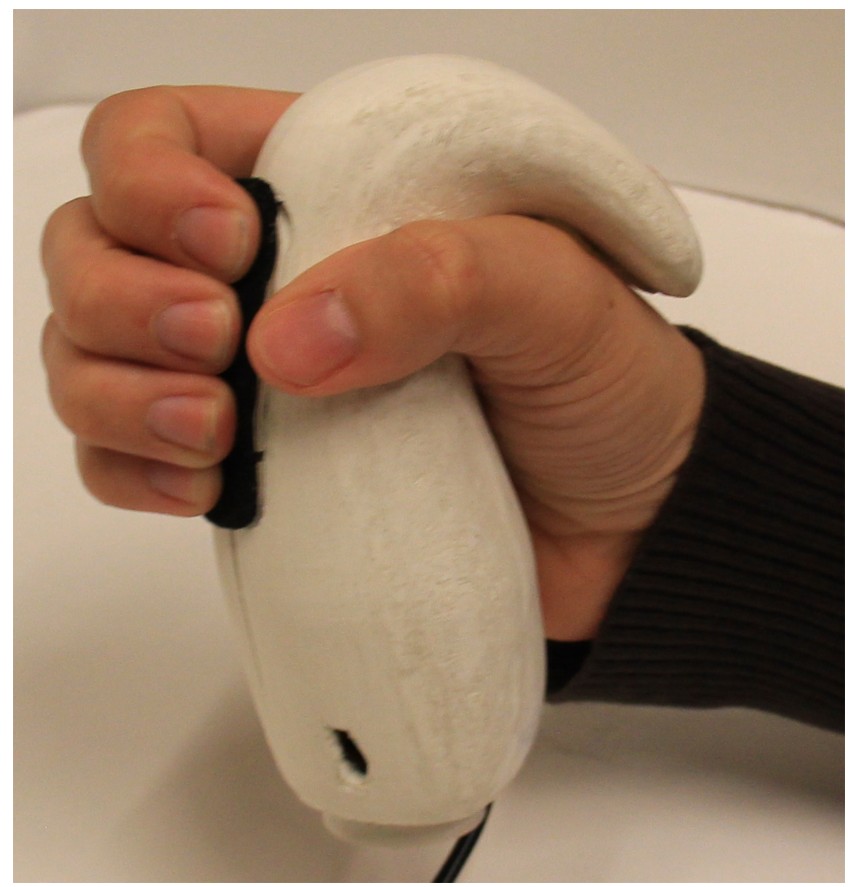

**Figure 1 A participant's hand holding a Frebble mediated touch device.**

## Mediated touch

Motivated by the fundamental importance of touch as a channel to convey human emotions, and the fact that even simple haptic stimulation can carry emotional information (*Salminen et al., 2008*), there has recently been an increasing interest in mediating touch for interpersonal communication in addition to vision and audition (*Toet et al., 2013*; *Van Erp & Toet, 2015*). Tactile or kinesthetic interfaces that enable haptic communication between people who are physically apart may thus provide the experiences of connection and engagement, with all the physical, emotional and intellectual feedback it supplies (*Cranny-Francis, 2011*). It has been shown that mediated handshaking can indeed enhance the feeling of social presence (*Nakanishi, Tanaka & Wada, 2014*). Also, haptic telecommunication can increase the quality of a shared experience and enhance the intimacy felt towards the other person (*Takahashi et al., 2011*) by intensifying the emotional displays from other (e.g., visual, auditory) modalities (*Knapp & Hall, 2010*) and by conveying specific emotions (*Hertenstein et al., 2006*). Recent technologies like HugMe (*Cha et al., 2009*), iFeel_ IM (*Tsetserukou et al., 2009*) and the Frebble (Fig. 1 ; see also *Toet et al., 2013* and http://myfrebble.com) therefore focus on the enhancement of social interactivity by enabling online communication programs (like MSN or Skype) to convey

affective and intimate haptic messages. It has also been suggested that this technology can be used to record affective touch patterns from loved ones for later replay (*Brown, 2015*). However, until now only few studies have actually investigated affect conveyance through mediated social touch (*Toet et al., 2013*). Although it appears that mediated touch can indeed to some extent convey emotions (*Bailenson et al., 2007*) and induce prosocial behavior (the Midas effect, *Haans & IJsselsteijn, 2009*), it is still not known to what extent it can also elicit affective experiences (*Gallace & Spence, 2010*; *Haans & IJsselsteijn, 2006*).

## Recovery from a sad experience

People feel a fundamental need to share their feelings (*Rimé, 2009*) and often seek haptic feedback (e.g., a hug, a pat on the arm or hand), especially after experiencing negative emotions since this type of feedback is highly effective in conveying support and providing comfort (*Dolin & Booth-Butterfield, 1993*). Responsive social touch not only positively affects the receiver but also the touching partner (*Debrot et al., 2013*; *Debrot et al., 2014*). A recent study found that even mediated touch can modulate emotion: mediated pressure to the forearm reduced heartbeat rates of participants after they had watched a sad video clip (*Cabibihan, Zheng & Cher, 2012*). Based on these findings we hypothesize that:

**H1**: Feedback via mediated hand touching enhances recovery from movie induced sadness to a larger extent than visual feedback.

## Intensified affective experience

People feel a need to communicate intense emotions with each other (*Rimé, 2009*). It has been found that people who experienced negative emotions (sadness, fear, disgust) after watching negative emotional movie clips shared more information and experiences with their friends than participants who watched neutral movie clips (*Luminet et al., 2000*). But people also like to share positive emotions like happiness and joy with each other, probably because sharing increases the possibility of personal rewards and consequently results in even more optimistic and positive feelings (*Bagozzi, Gopinath & Nyer, 1999*). Laughing together at a comedy show or sharing other moments of fun with each other improves and strengthens relationships and is an opportunity to create connections and bonds between people (*Klein, 1989*). Laughing is a natural social response that is contagious (*Rizzolatti et al., 1999*): people laugh more if others in their environment are also laughing (*Robinson & Smith-Lovin, 2001*). Touch can effectively intensify affective feelings and communication, even when it is mediated. For instance, mediated touch can enhance the affective quality of phone messages (*Salminen et al., 2012*) and remotely shared experiences (e.g., the hilariousness of a movie: *Takahashi et al., 2011*). A study on the added value of mediated touch enabled phone communication showed that couples are naturally inclined to use touch to express their amusement (*Park, Baek & Nam, 2013*). Given that amusement can be communicated by touching hands (*Hertenstein et al., 2009*), and that touch is capable to enhance affective feelings, our second hypothesis is therefore:

**H2**: Feedback via mediated hand touching enhances the experience of an amusing movie compared to visual feedback.

## Increase of trust

In a supportive setting interpersonal touch tends to increase trust, even among strangers (*Burgoon, Walther & Baesler, 1992*). It has been shown that mediated communication can also establish trust at a distance (*Bos et al., 2002*; *Zheng et al., 2002*). Although the effectiveness of mediated communication for establishing trust increases with media richness, trust is still highest when people can meet face-to-face (*Bos et al., 2002*; *Zheng et al., 2002*). This observation has simply been stated as "trust need touch" (*Handy, 1995*). The current study aims to investigate if mediated touch can also serve to establish trust. Our third hypothesis is therefore:

**H3:** Feedback through mediated hand touching increases trust towards another person compared to abstract visual feedback.

## Present study

To test the first two hypotheses (H1 and H2) participants first watched a sad movie clip to induce sadness, followed by a funny one to stimulate recovery from sadness. While watching the funny clip they signaled a hypothetical fellow participant (who was supposedly watching the same movies in an adjacent room) every time they were amused. These conditions were chosen since (1) people feel a fundamental need to share their feelings after watching a sad movie clip (*Luminet et al., 2000*), (2) they are inclined to use touch to express their amusement (*Park, Baek & Nam, 2013*), (3) feelings of amusement can be communicated by touching hands (*Hertenstein et al., 2009*), and (4) mediated tactile stimulation may enhance the experience of an amusing movie (*Takahashi et al., 2011*). In the experimental condition the participants responded by pressing a hand-held mediated touch device, which also provided haptic feedback (from the ostensible fellow participant) via a simulated hand squeeze. In the control condition they communicated by pressing a button and they received abstract visual feedback. Thus, participants gave haptic response in both conditions, and received haptic feedback in the experimental condition and abstract visual feedback in the control condition. Objective (physiological measurements) and subjective (questionnaires) measures were obtained to verify the hypotheses H1 and H2. To test whether mediated hand touching increased trust (H3) the participants played a trust game with their ostensible fellow participant at the end of the experiment.

## METHOD

### Participants

A total of $N = 39$ participants took part in this experiment. The data from two participants was excluded (one because of technical problems, and the other one was excluded because the participant provided no response during the experiment), resulting in a sample of $N = 37$. Of those, 23 were female and 14 were male, aged from 19 to 50 years ($M = 30.16$, $SD = 10.49$). The participants were recruited from the TNO database of volunteers and received 20 Euros for their participation. All participants gave their written consent. After the experiment the aim of the study was explained in a debriefing, with the exception
**Table 1 Sample size (*N*), mean (*M*) and standard deviations (*SD*) for both conditions.**

| | Age | | | | | |
| --- | --- | --- | --- | --- | --- | --- |
| | Men | | | Women | | |
| | *N* | *M* | *SD* | *N* | *M* | *SD* |
| Control condition | 7 | 29.57 | 11.55 | 11 | 28.00 | 10.46 |
| Experimental condition | 7 | 34.14 | 11.38 | 12 | 30.17 | 10.13 |

that the participants were not informed that the feedback they had received during the experiment had been scripted and was not sent by a fellow participant. This was done to prevent them from passing this information on to future participants.

None of the participants had critical heart diseases or used antidepressants that could affect heart rate. Participants were randomly assigned to either the control or the experimental condition. 18 participants performed in the control condition and 19 in the experimental condition. Table 1 shows how the participants were distributed over the experimental conditions with respect to sex and age.

## Experimental design

In a between-subjects design participants watched two emotional video clips while sharing their emotions in one of two ways with an ostensible fellow participant. In the experimental condition they signaled their own emotion by squeezing a mediated touch device, and they received haptic feedback about the feelings of their fellow participant in the form of hand squeezes presented via the same device. In the control condition they signaled their own emotions by squeezing the button on a joystick, and they received feedback in the form of an abstract visual cue.

Emotional video clips are known to effectively evoke and sustain affective experiences over longer time periods at both subjective and physiological levels (*Carvalho et al., 2012*). In this study the participants successively watched two emotional video clips: a sad one that served to elicit feelings of sadness followed by a funny one that was supposed to stimulate recovery from sadness. The participants were led to believe that they would be able to communicate with a fellow participant who would simultaneously be watching the same video clips in an adjacent room. In reality the feedback was scripted and no fellow participant was present. The two (control and experimental) conditions of this study differed only in the communication mode during the presentation of the second (funny) video clip (Fig. 2). In the control condition participants signaled which scenes they found funny by pressing a button on a joystick (haptic input), while visual feedback indicated whenever their ostensible fellow participant found a video segment funny. In the experimental condition participants held a mediated touch device in their hand which they could press (haptic input) to indicate which scenes they found funny. The same device squeezed their hand in return (haptic feedback) whenever their fellow participant supposedly found a video segment funny. Note that the haptic input modes in both conditions are equivalent since squeezing the joystick button required a similar action

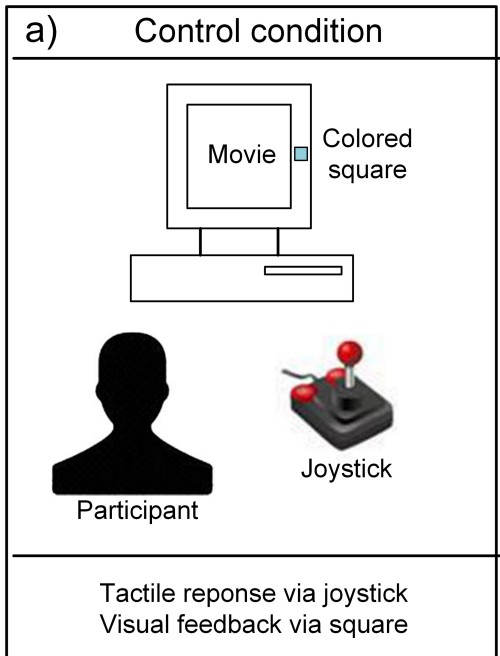
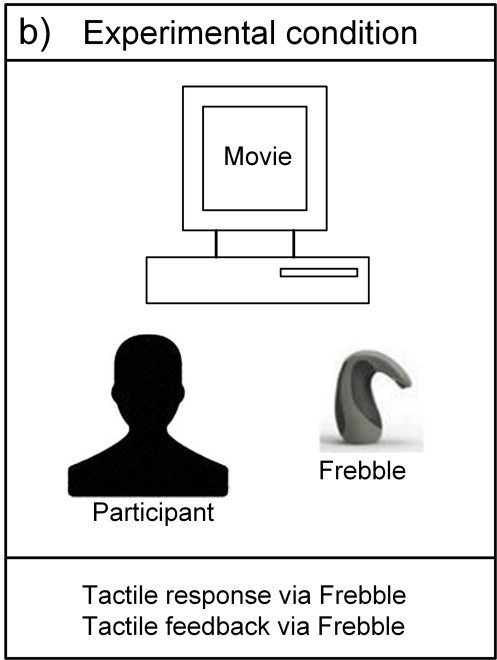

**Figure 2** Control (A) and experimental (B) conditions with response and feedback channels.

as squeezing the mediated touch device. The main difference between both conditions was the feedback mode, which was visual in the control condition and haptic in the experimental condition.

## Apparatus and stimuli

### Video clips and presentation

The clip that served to elicit sadness was a scene (with a duration of 686 s) from the 'The Champ' (*Lovell & Zeffirelli, 1979*) showing a boy who cries after his father's death. Several previous studies have shown that this scene successfully elicits feelings of sadness (*Ellard, Farchione & Barlow, 2012*; *Goldberg, Preminger & Malach, 2014*; *Gross & Levenson, 1995*; *Rottenberg, Ray & Gross, 2007*). The funny video (with a total duration of 595 s) was a compilation of 35 different scenes from funny home movies (www.youtube.com) showing hilarious scenes with animals and young children. In a pilot study this compilation was indeed rated on a 10-point semantic differential Likert Scale (1 = *not funny at all*; 10 = *very funny*) as being funny ($M = 7.86$; $SD = 1.45$; $N = 8$). The video clips used in this study had a resolution of $1,920 \times 1,080$ pixels. A PC equipped with a Sanyo PLC-WL2500A beamer (www.panasonic.com) projected the video onto an area of $190 \times 107$ cm$^2$ of the wall-mounted projection screen. Custom-made software was used both to present the scripted feedback and to play the movies. In the control condition a grey square ($60 \times 60$ pixels) in the middle of the right hand side of the screen turned blue to signal which scenes which the ostensible fellow participant found funny (visual feedback). The entire video presentation (including the instructions that appeared between the two video clips) lasted 21 min and 51 s.

### Joystick

In the control condition participants squeezed the rapid-fire trigger of a Logitech Extreme 3D Pro joystick (www.logitech.com) to signal to their fellow participant which video segments they found amusing.

### Frebble

In the experimental condition participants used a Frebble (Holland Haptics, Delft, The Netherlands; http://myfrebble.com) to communicate with their ostensible fellow participant. A Frebble is an ergonomically designed haptic device (Fig. 1, see also *Toet et al., 2013*) that comfortably fits in (and encloses the back of) the hand and simulates both the touch of holding and squeezing another person's hand—and having that feeling reciprocated (see also http://vimeo.com/86103101). Frebbles can communicate with one another via internet or Bluetooth. When a Frebble is squeezed, the corresponding gadget gently applies pressure to the back of a partner's hand, to simulate the feeling of holding hands. Two pressure sensors at the front of the device register squeezing while two vibration motors and a 'squeeze bar' provide the sensation of squeezing back. The bar is like a little lever that extends against the back of the hand. Frebbles are designed to enable mediated affective touch between physically separated partners. Users can communicate and receive haptic signals simultaneously. The pair of Frebbles used in this study consisted of one slave and one master, identical in size (12 cm high with a scope of 17 cm at the bottom and 11 cm at the top of the extension) and shape. Both Frebbles communicated with each other via Bluetooth. The pressure of the haptic signals provided by a Frebbble can vary from soft to hard. In this study the master Frebble was connected to the computer of the experimenter (via a USB connection) and sent scripted feedback to the slave Frebble (which was held by the participant) using a fixed pressure intensity, which resulted in a firm but comfortable squeeze (as established in a pilot test). Participants squeezed the Frebble to signal which video segments they found amusing, and the Frebble squeezed their hand in return whenever their fellow participant was supposedly amused.

### Scripted feedback

The scripted feedback for both conditions was determined from a pilot study in which 6 participants watched the funny video clip. Three participants used a joystick and three others used a Frebble to signal their response. They were instructed to press the button of the joystick or Frebble every time they were amused by the video clip. It was their own choice how often, for how long, and how firm they applied the pressure. The average of all participants' responses (duration and pressure level) was calculated and used as scripted feedback for the experiment. Although the fine nuances of the touch were lost by using a constant (average) pressure level, it makes the Frebble condition more comparable to the joystick condition (both are on or off at the same moments during the video clip). The scripted feedback was tested with two other participants to ensure that it was perceived as realistic and only occurred during video segments that could indeed be classified as amusing. The final feedback, which was used for the experiment, consisted of 57 presses with a total length of 191 s. The shortest response took 1 s and the longest 15 s.

During the experiment the same binary feedback track was used in both (visual and haptic feedback) conditions. In the control condition it resulted in visual feedback and in the experimental condition it led to a haptic (Frebble) feedback.

### Physiological measurements

Seven Ag–AgCL Flat electrodes were used to measure heart rate (HR) and skin conductance level (SCL). For the ground and reference measurement one electrode was placed on the neck of the participant and one behind each ear. The electrodes for the ECG signal were placed one at the right collarbone and one at the left floating rib. Two electrodes for the GSR signal were placed on the palm of the non-dominant hand of the participant (the hand not used to respond in the experiment), one right below the forefinger and the other one in the middle of the hand palm. The physiological signals were recorded with a computer using ActiView BioSemi software (www.biosemi.com) at a sampling rate of 512 Hz. The ECG signal was filtered using a 0.5–100 Hz bandpass filter, and afterwards by a 2.5 Hz high-pass 2-sided Butterworth filter. The GSC signal was filtered by a 30 Hz low-pass 2-sided Butterworth filter.

### Set-up conditions

The experiments were performed in a small, well-ventilated, and sound-proof windowless room. The lights were dimmed during the video presentation. The participants were seated comfortably on a coach facing a wall mounted projection screen from a mean distance of 3.20 m. To enable later analysis of their facial expressions participants were recorded on video during the experiment using a Panasonic WV-BP330/GE camera (www.panasonic.com) in combination with an infrared light source.

## Measures

### Objective measures

Heart rate and skin conductance level were used to quantify a participant's physical reactions to the video clips (*Davydov, Zech & Luminet, 2011*; *Kreibig, 2010*). Both heart rate (*Mandryk, Inkpen & Calvert, 2006*) and skin conductance level (*Lang, 1995*) are known to reflect emotional activity (*Brouwer et al., 2013*). Sad film clips typically evoke a decreased skin conductance level and a decreased heart rate (i.e., when participants do not cry in response to watching the film: *Kreibig, 2010*). In contrast, amusing film clips typically evoke an increase in skin conductance level, often accompanied by an increase in heart rate (although the heart rate response is not unequivocal: *Kreibig, 2010*). The physiological data were processed and analyzed in Matlab 8.0 (www.mathworks.com). The mean value of a participant's heart rate and skin conductance level were computed over the following three time intervals (later referred to as the *time of measurement*): (1) the preparation period before watching the movie clips (this served as a personal baseline), (2) the last four minutes of 'The Champ' (just after the father's death, which is the saddest part of the movie) and (3) the first 90 s of the funny video clip (the period during which recovery effects are most likely to occur).

In a previous study on empathic touch by relational agents *Bickmore et al. (2010)* observed that the number of hand squeezes and squeeze duration were associated with
affect valence when mediated touch was used as the only feedback mode. In the present study we therefore adopted the number and duration of Frebble responses and the total duration of smiling (estimated from the analysis of the video recordings of a participant's facial expressions) during the presentation of the funny video clip as additional objective measures of the participant's affective response to the clip.

The duration of a participant's smile while watching the funny video clip was measured from the video recording of the participant's facial expressions using The Observer XT 11.5 software tool (www.noldus.com). Therefore the experimenter watched the video recording of each participant and judged once per second whether the participant was smiling or not. A smile was recognized according to the definition of a Duchenne smile (*Soussignan, 2002*). To assess the reliability of this annotation method two additional observers also annotated five randomly selected videos and their inter-rater reliability (Person's $r$) with the experimenter was calculated. The inter-rater reliability between the observers and the experimenter was high: $r = .89$ for both observers.

The amount of money set in during the trust game was used as an objective measurement of trust towards the fellow participant.

### Subjective measures

During the experiment each participant filled out three questionnaires: the first one at the start of the experiment (70 questions and statements), the second one after watching the movie clips (32 questions and statements) and the third one at the end of the experiment when the experimenter pretended to play a trust game with the fellow participant (16 questions and statements).

## Questionnaire 1

The first questionnaire contained demographic questions about gender, age, heart or other diseases, and use of medicines. To explain possible outliers in the physiological data the participants were also asked how much time had passed since they smoked, drank coffee or alcohol, or had any physical exercise before the experiment. Also, questions were asked about their current mood, personality and touch receptivity. These questions served to check whether the groups of participants in both experimental conditions were comparable in terms of current mood, personality and touch receptivity.

The *current mood* of the participants was measured with the 'Brief Mood Introspection Scale' (BMIS: *Mayer & Gaschke, 1988*) which consisted of 16 adjectives (such as *lively*, *happy*, *sad* and *tired*) that could be rated on a 4-point Likert Scale (1 = *definitely do not feel*; 4 = *definitely feel*). The reliability (Cronbach's alpha) of the BMIS for the sample in this study was $\alpha = .82$. The 'Self-Assessment Manikin' (SAM: *Bradley & Lang, 1994*) was also used to assess the current mood. The SAM is a nine-point pictorial rating scale that measures pleasure, arousal and dominance. The SAM provides a simple, fast, and non-linguistic way of assessing a person's mental state along the principal emotional dimensions and is highly suitable to measure transient (short term) emotional states. The reliability for this scale was $\alpha = .63$.

Three personality traits (*Extraversion*, *Openness* and *Agreeableness*) were measured with the Dutch version (*Van Heck et al., 1994*) of the original 'Big 5' questionnaire (*Goldberg, 1992*). For each personality trait 10 statements had to be answered on a 5-point Likert Scale (1 = *strongly disagree*; 5 = *strongly agree*). An example question for measuring Extraversion was: "*I feel comfortable around people*". The reliabilities were $\alpha = .88$ for *Extraversion*, $\alpha = .85$ for *Openness* and $\alpha = .56$ for *Agreeableness*.

Eight items from the Touch Receptivity Questionnaire (the same items used by *Bickmore et al., 2010*, for example: "*I like people who shake hands with me.*") were used to measure how comfortable participants were with being touched by someone else. The eight items were translated into Dutch and were answered on a 7-point Likert Scale (1 = *disagree completely*, 7 = *agree completely*). They had a reliability of $\alpha = .76$.

## Questionnaire 2

After watching both movie clips the participants were asked to fill out a second questionnaire to measure their affective experience. In addition to questions about the movie itself this questionnaire also presented the BMIS and SAM again (for the second time). This was done to identify possible changes in mood after watching the movie clips.

Participants were asked to indicate how entertaining they found the funny movie clip on a 10-point semantic differential Likert Scale (1 = *not funny at all*; 10 = *very funny*). In addition they were asked to rate their overall feeling while watching the funny clip by rating 10 adjectives (*interested, joyful, sad, angry,fearful, terrified, contempt, disgusted, surprised, happy*) on a 6-point Likert Scale (1 = *not at all*; 6 = *very intense*; e.g., *Davydov, Zech & Luminet, 2011*; *Schaefer et al., 2003*). The reliability of these items was $\alpha = .91$. Together these questionnaires provided an impression of the overall affective experience of the funny movie clip.

Participants were also asked whether they had ever seen the first movie clip ('The Champ') before and rated on a 10 point scale how sad they experienced it (1 = *not at all*, 10 = *very intense*).

## Questionnaire 3

A third questionnaire served measured the participant's impression of the (ostensible) fellow participant.

Affective trust is the confidence one places in a partner on the basis of feelings generated by the level of care and concern the partner demonstrates (*Johnson & Grayson, 2005*). In this study affective trust towards the fellow participant was measured with one item ("*I felt that the other person understood me.*") that was rated on a 7-point Likert Scale (1 = *completely disagree*; 7 = *completely agree*).

A single item ("*I felt that the other person had the same humor.*") rated on a 7-point Likert Scale (1 = *completely disagree*; 7 = *completely agree*) measured whether the participant thought that the fellow participant appreciated the same kind of humor.

Attachment was measured with an adapted version of a questionnaire that was originally developed to measure consumer-product attachment (*Schifferstein & Zwartkruis-Pelgrim, 2008*). This questionnaire contained five items ("*I felt emotionally connected with my*

*fellow participant.*", "*The fellow participant was dear to me*"., "*I had a bond with my fellow participant.*", "*The fellow participant had no*special*meaning for me.*", "*I had no feelings for my fellow participant.*") that could be rated on a 7-point Likert Scale (1 = *completely disagree*, 7 = *completely agree*). This questionnaire had a reliability of $\alpha = .89$.

Perceived trustworthiness of the fellow participant was measured on the dimensions trust, immediacy, reliability, and credibility (*Rubin et al., 2009*) using eight 7-Point bipolar semantic differential Likert scales (*cold–warm, familiar–unfamiliar, friendly-unfriendly, distant–close, kind–cruel, active–passive, reliable–unreliable, direct–indirect*). The reliability of this questionnaire was $\alpha = .73$.

The 'Inclusion of Other in the Self' Scale (IOS: *Aron, Aron & Smollan, 1992*) is a single-item, pictorial measurement of closeness between two persons. It shows seven pairs of circles (one circle representing the Self and the other one representing the partner) with different degrees of overlap representing different degrees of interpersonal interconnectedness. The degree of overlap between the circles increases linearly from non-overlapping (1; representing *no interpersonal connection*) to almost completely overlapping (7; representing *a strong interpersonal connection*). The participants were asked to mark the picture which best described the relationship they experienced to their fellow participant.

## Trust game

To test whether mediated hand touching can increase trust towards another person (H3) the participants were invited to play a trust game with their ostensible fellow participant at the end of the experiment. After watching both movies and filling out the second questionnaire the experimenter thanked the participants and handed them their show-up fee. At that time the experimenter informed the participants that they were given a chance to increase their fee by participating in a game. The participants were free to decide whether or not to participate in the game and how much of their fee they would send to their fellow participant. They were told that their fellow participant would receive twice the amount of money they sent and that he/she could decide whether to keep the money or to return a certain amount to the sender. In this game, sending money to the other player is risky but can also lead to an expanded profit, if returned by the other player. The amount of money that the sender decides to transfer to the recipient can be seen as a (behavioral) measure of the sender's trust towards the other person. This so called 'trust game' is a well-established method to measure the degree of trust between persons (*Berg, Dickhaut & McCabe, 1995*; *Glaeser et al., 2000*; *Lazzarini et al., 2004*). In the current study there was actually no second player involved, and the experimenter returned all participants the same amount they gave away.

## Statistical analysis

The statistical data analysis was done with IBM SPSS 20.0 (www.ibm.com). The dependent variables of this study were: the physiological measurement of heart rate and skin conductance level, the variables of the questionnaires, the Frebble and joystick response behavior (number and duration of responses) while watching the funny movie clips, the
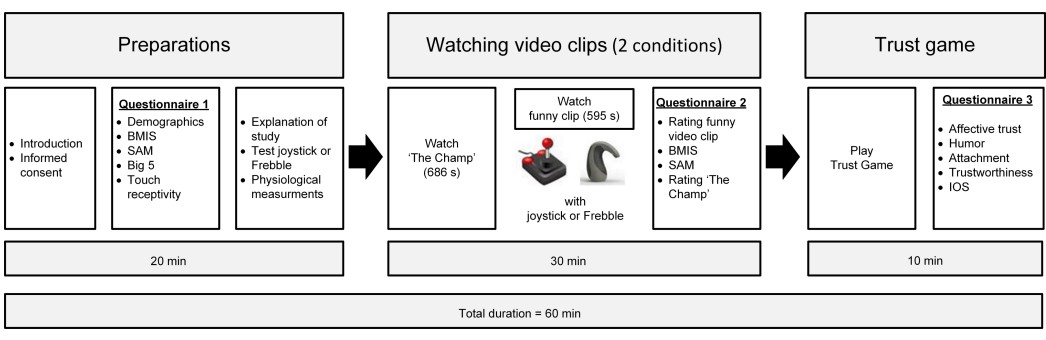

**Figure 3** Timeline of the experimental procedure.

amount of money set in at the trust game, and the facial expression data (smile duration). Independent sample *t*-tests and mixed design analyses of variance (ANOVAs) were conducted for the normally distributed dependent variables. A Wilcoxon–Mann–Whitney test was conducted for the dependent variables of Frebble press behavior (duration and number of response) and the smile duration, because these data were not normally distributed. Three participants incorrectly used the "*movie rating*" answer categories. It was therefore decided to exclude their answers from the analysis of the dependent variable '*movie rating adjectives*'. All analyses of the physiological measurements were done both with difference scores (difference between the individual baseline and the measured value during the movie clips) as well as with mean scores (mean value of all participants measured during the movie clips). Because both analyses showed no difference in outcomes only the mean values of heart rate and skin conductance level are reported this study.

## Procedure

The timeline of the experimental procedure is shown in Fig. 3. After having been picked up and greeted in the entrance hall by the experimenter the participant was guided to the test room, sat down on a sofa, and read and signed an informed consent form. The participant was then told that the experimenter had to return to the entrance hall to pick up a second participant, who was intentionally invited 5 min later to prevent both participants from meeting each other. The participant was informed that it was necessary to avoid a physical encounter with the other participant since a first impression of the other participant based on factors like gender and attractiveness might influence the outcome of the study. The experimenter asked the participant to fill out the first questionnaire in the meantime and to wait until she would return. In fact the experimenter waited outside the test room and returned 15 min later when the participant had finished the questionnaire. This procedure served to enhance the illusion that there was indeed a fellow participant present in a room next to the test room. The experimenter explained that the purpose of the study was to investigate people's emotions while watching emotional video clips. Therefore the participant would watch two video clips that were also simultaneously being watched by the fellow participant in the adjacent room. The participant was asked to simply watch the first video clip, and to signal the fellow participant in the other room whenever an amusing

episode occurred in the second video clip by pressing the Frebble (experimental condition) or by pressing the button of the joystick (control condition). The participant was told that the fellow participant would do the same. The feedback of the fellow participant was either in the form of a simulated hand press via the feedback mechanisms of the Frebble (experimental condition—haptic feedback) or shown on the projection screen in the form of a grey square that turned blue whenever the fellow participant signaled the occurrence of an amusing episode (control condition—visual feedback). In the experimental condition the participants were led to believe that their fellow participant also held a Frebble device and would experience their responses as hand squeezes. In the control condition they were told that their fellow participant would see their response as a color change of a square shown on the projection screen. The way to use the Frebble or the joystick to send information and the meaning of the feedback (either presented via the Frebble or via the colored square) were explained and the participant was given the opportunity to experience it and to ask questions. If there were no further questions the electrodes to measure heart rate and skin conductance level were attached to the participant. Then the participant was asked to sit calmly for a few minutes before the video clips would start. The participant was informed that this was necessary because the experimenter also needed to explain the experiment to the other participant. The participant was left alone and the video clips would start automatically after eight minutes. The sound of the video clips was played through headphones to exclude any noise from the Frebble's actuators. After the first video clip ('The Champ') a text appeared on the screen instructing the participant to use the joystick or Frebble to communicate with the fellow participant during the presentation of the second video clip. After the second video clip the participant filled out a second questionnaire. After a short waiting period the experimenter returned in the room, removed the sensors from the participant's body, placed the participant's fee on the table (20 coins of 1 Euro) and asked if the participant would like to play a game with the fellow participant. The experimenter explained the game and then played the first round with the participant. Next she pretended to play the second round with the fellow participant in the adjacent room. While the experimenter was outside the room pretending to play the game with the fellow participant, the participant completed the last (third) questionnaire. The experimenter returned after a few minutes with the same amount of Euros the participant had previously sent to the fellow participant. All of the 37 participants agreed to play the game. The total duration of the experiment was about 60 min for each participant. The experimental protocol was reviewed and approved by the TNO internal review board on experiments with human participants (TNO Soesterberg, The Netherlands), and was in accordance with the Helsinki Declaration of 1975, as revised in 2000 (*World Medical Association, 2000*).

## RESULTS

An independent $t$-test on the scores on the Touch Receptivity Questionnaire, the BMIS, the three dimensions of the SAM (pleasure, arousal and dominance) and the three personality traits of the BIG 5 (*Extraversion*, *Openness* and *Agreeableness*) from Questionnaire 1

**Table 2 Sample size (N), means (M), standard deviations (SD) for the joystick and Frebble condition.** Also shown are the *p*-values of the independent samples *t*-test for each variable between both conditions.

| | Joystick | | | Frebble | | | t-test |
|---|---|---|---|---|---|---|---|
| | N | M | SD | N | M | SD | p |
| Brief Mood Introspection Scale (BMIS) | 18 | 3.14 | 0.23 | 19 | 3.13 | 0.35 | .915 |
| Self-Assessment Manikin (SAM)—pleasure | 17 | 6.94 | 0.75 | 19 | 6.68 | 1.34 | .476 |
| Self-Assessment Manikin (SAM)—arousal | 18 | 3.50 | 1.76 | 19 | 2.58 | 1.54 | .098 |
| Self-Assessment Manikin (SAM)—dominance | 17 | 4.94 | 0.83 | 19 | 5.42 | 1.84 | .313 |
| BIG 5—extraversion | 18 | 3.17 | 0.68 | 19 | 3.56 | 0.63 | .078 |
| BIG 5—openness | 18 | 3.49 | 0.56 | 19 | 3.61 | 0.61 | .567 |
| BIG 5—agreeableness | 18 | 3.47 | 0.46 | 19 | 3.48 | 0.31 | .958 |
| Touch receptivity | 18 | 4.76 | 1.01 | 19 | 4.57 | 0.95 | .565 |

showed no significant differences between both experimental groups (all *p*-values > .05; see Table 2). Hence, both groups are indeed equivalent, which makes any differences between their responses to Questionnaires 2 and 3 most likely the result of the different experimental conditions.

## Emotion elicitation

A prerequisite for testing the three hypotheses was that the video clip from 'The Champ' would elicit sad emotions whereas the funny video clips would cheer people up. Objective and subjective measurements were therefore analyzed to investigate if these manipulations had been successful.

## Objective measurements

The mean value of a participant's heart rate and skin conductance level were computed over the following three time intervals: (1) the preparation period before watching the movie clips (this served as a personal baseline), (2) the last four minutes of 'The Champ' (the part after the father's death) and (3) the first 90 s of the funny video clip. Figure 4 shows the mean values of these measures over all participants. On first inspection the ranking of the mean heart rate and mean skin conductance levels during the three time intervals appears to agree with our expectations: the mean levels of these parameters during the sad and funny movie periods were respectively lower and higher than their corresponding values during the baseline period.

A repeated measures ANOVA was conducted to investigate whether the mean heart rate differed significantly between the three times of measurement. The results showed that the heart rate was indeed significantly affected by the time of measurement, $F(2, 72) = 4.58$, $p = .013, \eta^4 = .113$. A Bonferroni post-hoc comparison of the three intervals indicated that the heart rate during the last minutes of 'The Champ' ($M = 68.20$, 95% CI [65.77–70.63]) was significantly lower than during the first 90 s of the funny video clips ($M = 69.92$, 95% CI [67.08–72.75]), $p = .020$. The baseline ($M = 69.03$, 95% CI [66.41–71.64]) did not differ significantly from 'The Champ' ($p = .30$) and the funny video clips ($p = .46$).

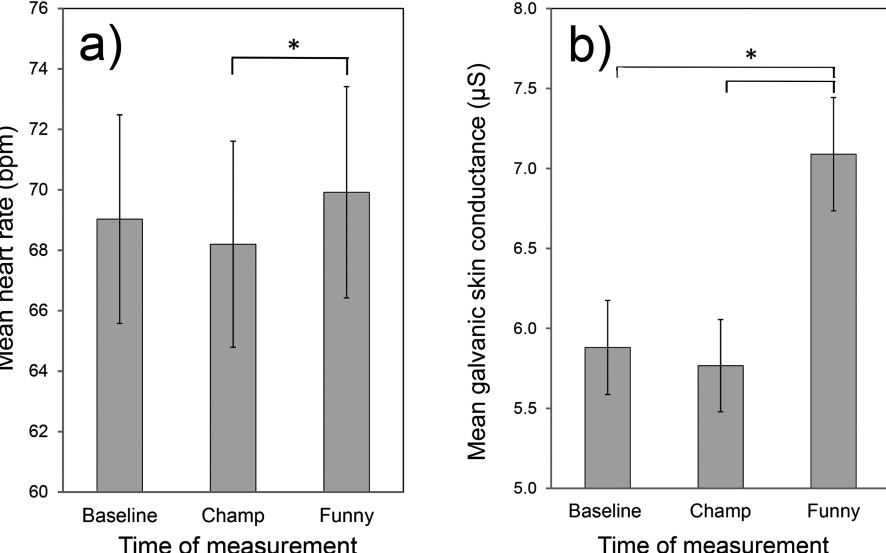

**Figure 4 Mean (over all participants) heart rate (A) and galvanic skin conductance (B) at different times of measurement.** Measurements were obtained before seeing the videos (baseline), during the last part of the 'The Champ', and during the funny video clip. The error bars represent confidence intervals, while * indicates that Bonferroni post-hoc comparison was significant ($p < 0.05$).

A repeated measures ANOVA between the three times of measurement was also conducted for the mean skin conductance level. The analysis showed that the mean skin conductance level was significantly affected by the time of measurement, $F(2, 72) = 13.73$, $p < .001$, $\eta^2 = .276$. A Bonferroni post-hoc comparison indicated that the skin conductance level was significantly lower during the baseline period ($M = 5.88$, 95% CI [5.05–6.71]) than during the presentation of the funny video clip ($M = 7.09$, 95% CI [6.27–7.91]), $p = .001$. The skin conductance level during the last four minutes of 'The Champ" ($M = 5.77$, 95% CI [4.84–6.70]) was also significantly lower than the skin conductance level during the funny video clip, $p < .001$. There was no significant difference between the skin conductance level during the baseline period and during 'The Champ' ($p = 1.00$).

## Subjective measurements

'The Champ' scored on average 7.09 ($SD = 1.79$) on a scale from 0 (*not sad at all*) up to 10 (*very sad*) on the self-reporting questionnaire. The funny video clip scored on average 6.84 ($SD = 1.42$) on a scale from 0 (*not funny at all*) to 10 (*very funny*). In addition, in response to the 10 questions about their feelings while watching the funny movie clips it appears that participants on average felt quite positive: $M = 5.03$ ($SD = 0.57$) on a 6 point Likert scale.

## Conclusion

The subjective measurements indicated that both movies served their purpose: 'The Champ' indeed elicited sad emotions while the funny video clip appeared to evoke more positive feelings. Although the ranking of the objective measurements (mean heart rate and mean skin conductance levels) agrees with the subjective measurements,

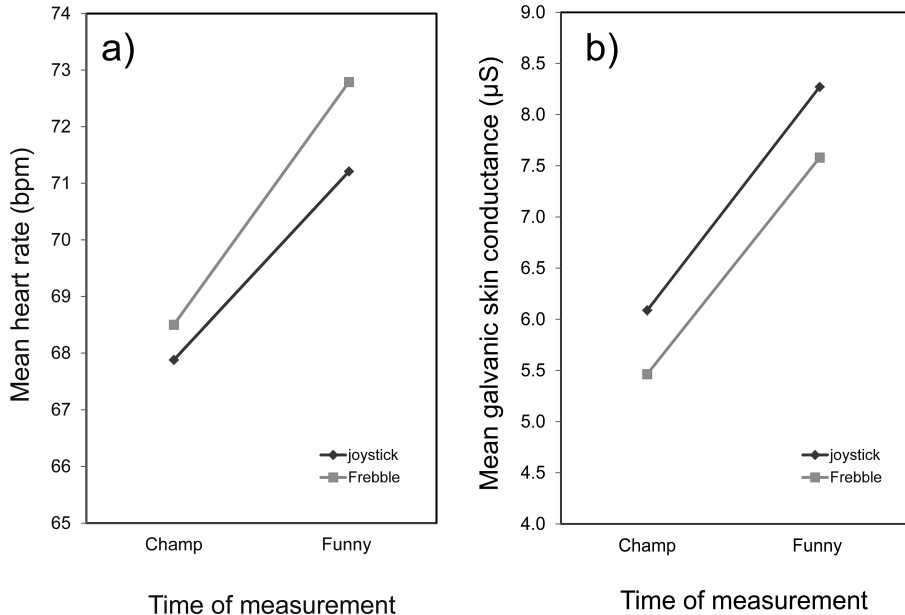

**Figure 5 Mean heart rate (A) and mean galvanic skin conductance (B).** Measurements were obtained during 'The Champ' and at the beginning of the funny movie clips in respectively the joystick and Frebble conditions.

this conclusion was only confirmed for the funny video clip (both heart rate and skin conduction level were significantly higher than the baseline value during the first 90 s of the funny movie), but not for the sad video clip (both heart rate and skin conduction level while watching 'The Champ' were not significantly different from the baseline value).

## Hypothesis I: Recovery from a sad experience

To test whether mediated touch communication enhances recovery from movie induced sadness to a larger extent than visual feedback (H1) the physiological measurements (heart rate and skin conductance level) in the control and Frebble conditions were compared between 'The Champ' (last four minutes after the father's death) and the recovery period during the funny video clip (the first 90 s) using a mixed-model ANOVA.

### Heart rate

A mixed-model ANOVA revealed that there was a main effect of *time of measurement* between 'The Champ' and the funny video clip, $F(1, 35) = 21.093$, $p < .001$, $\eta^2 = .376$ (Fig. 5A). However there was no significant main effect of *condition*, $F(1, 35) = 0.172$, $p = .680$, $\eta^2 = .122$, and no interaction effect between time *of measurement* and *condition*, $F(1, 35) = 0.332$, $p = .568$, $\eta^2 = .014$. This indicates that the participants' heart rate did not differ significantly between the control and Frebble conditions over the period defined by the presentation of 'The Champ' and the first 90 s of the funny video clip.

### Skin conductance

A mixed-model ANOVA again revealed that there was a main effect of *time of measurement* between 'The Champ' and the funny movie clips, $F(1, 35) = 329.62$, $p < .001$, $\eta^2 = .904$

(Fig. 5B). However, there was no significant main effect of *condition*, $F(1, 35) = 0.62$, $p = .434$, $\eta^2 = .072$. and no interaction effect between *time of measurement* and *condition*, $F(1, 35) = 0.627$, $p = .434$, $\eta^2 = .015$. This also indicates that there was no significant difference between the joystick and Frebble condition during both time intervals.

### Conclusion

The variation in mean heart rate and mean skin conductance level over both measurement intervals (the last four minutes of 'The Champ' and the first 90 s of the funny video clip) indicates that the participants did indeed recover from the sad movie over the associated time period. However, mean heart rate or mean skin conductance level did not differ significantly between both conditions over these measurement intervals. Hence, feedback via mediated hand touching did not enhance the recovery from movie induced sadness to a larger extent than visual feedback (H1). Hypothesis 1 was therefore not confirmed.

## Hypothesis 2: Intensified affective experience

To test whether feedback via mediated hand touching enhances the experience of an amusing movie relative to visual feedback (H2) we compared the objective (press and smiling behavior) and subjective (questions about the affective experience in Questionnaire 2) measurements in the control and Frebble conditions.

### Objective measurements

The objective measurements of the press behavior (duration and number of responses) and the time the participants were smiling were not completely normally distributed. Therefore, a non-parametric Kolmogorov–Smirnov test with ranked scores and a Wilcoxon–Mann–Whitney test were used to analyze these three dependent variables. Because the results were almost identical only the results of the Wilcoxon–Mann–Whitney test will be reported.

The time during which the joystick was pressed while watching the funny video clip (*Press duration*) in the control condition ($Mdn = 59.56$) did not differ significantly from the time during which the Frebble was pressed in the experimental condition ($Mdn = 86.61$), $U = 227.00$, $z = -1.702$, $p = .092$, ns. Also, the number of presses in the control condition ($Mdn = 49.00$) did not differ from the number in the Frebble condition ($Mdn = 52.00$), $U = 151.50$, $z = -0.593$, $p = .563$, ns. The annotation of the facial expression indicated that the time the participants were smiling while watching the funny video clip also did not differ significantly between both conditions (joystick: $Mdn = 166.00$; Frebble: $Mdn = 225.00$), $U = 138.50$, $z = -0.988$, $p = .331$ ns.

*Press duration* correlated significantly with: *Openness* ($r = .34$, $p < .05$), the amount of money set in the trust game ($r = .35$, $p < .05$), and with affective trust towards the fellow participant ("*I felt that the other person understood me*": $r = .43$, $p < .01$). *Number of presses* is significantly correlated with the amusement rating of the funny movie clip ($r = .43$, $p < .01$), the score on the second BMIS (after watching the funny movie: $r = .35$, $p < .05$), the *Pleasure* score on the SAM ($r = .42$, $p < .01$), affective trust towards the fellow participant ($r = .52$, $p < .01$), and the assessment of the fellow particpant's sense of humor

("*I felt that the other person had the same humor*": $r = .57, p < .01$). *Number of presses* and *press duration* were also significantly correlated ($r = .41, p < .05$).

Finally, an independent samples *t*-test was used to investigate whether the mean heart rate and skin conductance measurements (both registered while watching the funny video clip) differed significantly between both conditions. For mean heart rate there was no significant difference between the control ($M = 69.55; SD = 7.76$) and the Frebble condition ($M = 70.27; SD = 9.37$), $t(35) = -0.253, p = .801$, ns. For skin conductance the control ($M = 7.60; SD = 2.69$) and Frebble ($M = 6.61; SD = 2.16$) conditions showed no significant difference either, $t(35) = 1.240, p = .223$, ns.

### Subjective measurements

An independent samples *t*-test was used to investigate whether the participants scored differently in the control and Frebble conditions on the questions about their affective experience of the funny movie clips. For the rating score (from 0 to 10) about how amusing the participants found the movie there was no significant difference between the control ($M = 6.67; SD = 1.65$) and the Frebble conditions ($M = 7.00; SD = 1.20$), $t(35) = -0.707$, $p = .485$, ns. For the rating of the movie using adjectives there was also no significant difference (control: $M = 5.09; SD = 0.65$; Frebble: $M = 4.98; SD = 0.51$), $t(32) = 0.526$, $p = .603$, ns.

### Conclusion

Neither the objective nor the subjective measurements showed any significant difference between the affective experience of the funny movie in both experimental conditions. Hypothesis 2 (feedback via mediated hand touching enhances the experience of an amusing movie compared to visual feedback) was therefore not confirmed.

## Hypothesis 3: Increase of trust

To test whether feedback through mediated hand touching increases trust towards another person relative to abstract visual feedback (H3) we compared the objective (amount of money bet during the trust game) and the subjective (questions concerning the impression of the other participant in Questionnaire 3) measurements in the control and Frebble conditions.

### Objective measurements

The amount of money bet during the trust game was used to measure trust towards the other person. An independent *t*-test between the control ($M = 6.83, SD = 5.76$) and the Frebble condition ($M = 6.21, SD = 5.91$) showed that there was no significant difference between the amount of money set in in both conditions; $t(35) = 0.324, p = .748$, ns.

### Subjective measurements

At the end of the experiment questions were also asked concerning the impression of the other participant. None of these items showed a significant difference between the conditions (all *p*-values $> .05$). The results of the independent *t*-tests are shown in Table 3.

**Table 3 Sample size (N), means (M), standard deviations (SD) for the variables concerning the other person, split up for the control and Frebble condition.** Also shown are the *p*-values of the independent samples *t*-test for each variable between both conditions.

| | Joystick | | | Frebble | | | *t*-test |
|---|---|---|---|---|---|---|---|
| | **N** | **M** | **SD** | **N** | **M** | **SD** | **p** |
| Affective trust | 18 | 3.44 | 1.38 | 19 | 3.84 | 1.61 | .426 |
| Humor | 18 | 3.83 | 1.89 | 19 | 3.58 | 1.58 | .658 |
| Attachment | 18 | 3.03 | 1.21 | 19 | 3.31 | 1.54 | .557 |
| Impression of the other person | 18 | 4.43 | 0.80 | 19 | 4.45 | 0.73 | .926 |
| Inclusion of Other in the Seelf (IOS) | 18 | 2.33 | 0.97 | 19 | 2.58 | 1.35 | .531 |

### *Conclusion*

The results of both the objective and subjective measurements indicate that compared to visual feedback, mediated haptic communication leads neither to more trust nor to a different impression of the other person. Hypothesis 3 (feedback through mediated hand touching increases trust towards another person compared to abstract visual feedback) was therefore not confirmed.

## Sample characteristics

The current mood (BMIS) and emotional state (SAM) of the participants were again measured after watching both video clips. To test whether mood and emotional state were affected differently in both conditions, for all four dependent variables a mixed-design ANOVA with a within-subjects independent variable of *time of measurement* (before and afterwards) and a between-subjects independent variable of *condition* (control vs. Frebble) was conducted.

For the BMIS there were no main effects of *time of measurement*, $F(1, 35) = 2.164$, $p = .150$, ns and condition, $F(1, 35) = 0.518$, $p = .518$, ns, nor was there an interaction between BMIS and condition, $F(1, 35) = 0.102$, $p = .752$, ns.

For the *Pleasure* measurement (SAM) there were no effects of *time of measurement*, $F(1, 34) = 0.003$, $p = .956$, ns and condition, $F(1, 34) = 0.007$, $p = .934$, ns. There was also no interaction between *time of measurement* and *condition*, $F(1, 34) = 1.007$, $p = .323$, ns.

The *Arousal* measurement (SAM) showed a main effect of *time of measurement*, $F(1, 35) = 4.513$, $p = .041$, $\eta^2 = .114$ and a significant interaction between *time of measurement* and *condition*, $F(1, 35) = 7.674$, $p = .009$, $\eta^2 = .180$ (Fig. 6). However, there was no main effect for *condition*, $F(1, 35) = 0.151$, $p = .700$, ns (Fig. 6).

Main effects of *time of measurement*, dominance (SAM), $F(1, 34) = 1.575$, $p = .218$, ns and *condition*, $F(1, 34) = 2.532$, $p = .121$, ns as well as the interaction between the *dominance* and *condition*, $F(1, 34) = 0.827$, $p = .370$, ns were not statistically significant.

## Explorative analyses

Besides testing the hypotheses of the present study there were also data collected concerning age, gender, mood (BMIS), emotions (SAM), Touch Receptivity, and personality traits

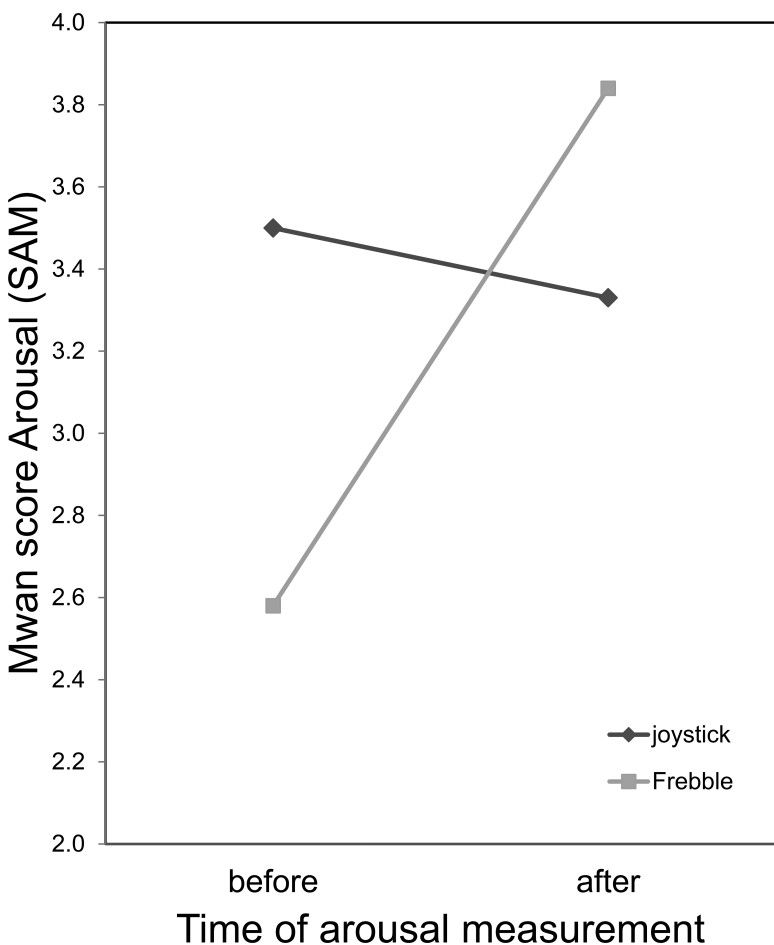

**Figure 6** Mean arousal scores (SAM) before and after the experiment, for the joystick and Frebble conditions.

(BIG 5). To investigate whether these factors may have an effect on the dependent variables some explorative analyzes were conducted. Therefore separate analyses of covariance for each dependent variable were conducted with each of the listed factors (age, gender, mood and emotions) as a covariate. The results showed no significant results. None of the factors influenced the outcome of the dependent variables as a covariate.

In addition, possible effects on the dependent variables were investigated by a two-way ANOVA having two levels for each factor (Table 4) and two levels of conditions (control vs. Frebble). Therefore each factor was subdivided in two levels based on the median score. The factors: *mood* (BMIS), *emotions* (all three SAM measurements), *Openness* and *Agreeableness* were excluded from this analysis because the scores were extremely right skewed with less than 15% of the scores being lower than the mean score. The independent variables used for the analyses with their two levels are shown in Table 4. All effects were statistically significant at the .05 significance level. It is chosen to only report significant results.

**Table 4** Sample size (*N*) and values for both levels of each factor based on the mean scores.

| | Level 1 | | Level 2 | |
| --- | --- | --- | --- | --- |
| | **N** | **Values** | **N** | **Values** |
| Age (in years) | 25 | Young (18–35) | 12 | Old (36–50) |
| Gender | 14 | Men | 23 | Women |
| Touch receptivity | 18 | Low (1–4, 5) | 19 | High (4, 6–7) |
| Extraversion | 11 | Low (1–3) | 26 | High (3, 1–5) |

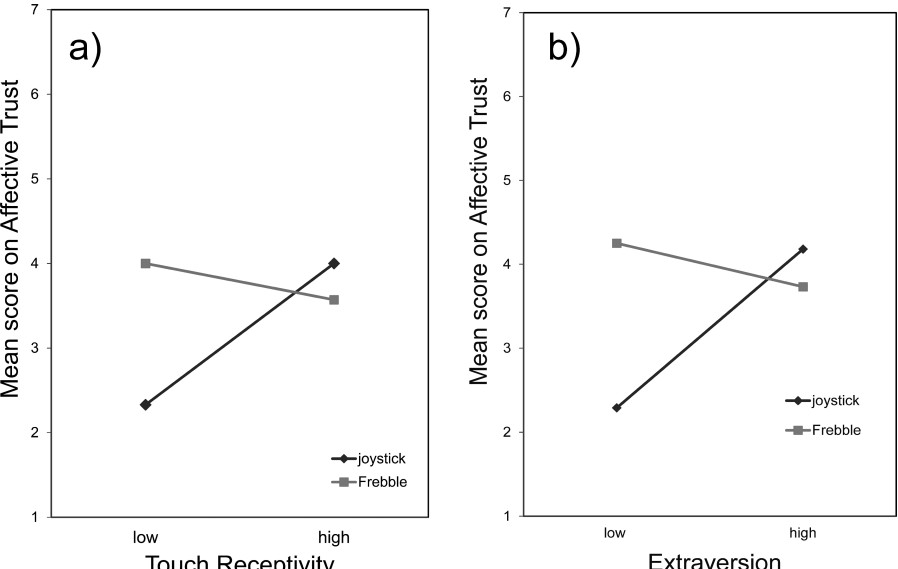

**Figure 7** Mean scores on affective trust in the joystick and Frebble conditions for the factors Touch Receptivity (A) and Extraversion (B).

There were no main effects of *Touch Receptivity*, $F(1, 33) = 1.584$, $p = .217$, ns or *condition*, $F(1, 33) = 1.584$, $p = .217$, ns, on the dependent variable of *Affective Trust*. However, there was a significant interaction effect between *Touch Receptivity* and *condition* on *Affective Trust*, $F(1, 33) = 4.537$, $p = .041$, $\eta^2 = .121$. Specifically, participants who scored low on *Touch Receptivity* reported higher *Affective Trust* towards their fellow participant when using a Frebble ($M = 4.00$, $SD = 0.41$) than when using a joystick ($M = 2.33$, $SD = 0.58$; Fig. 7A).

There were no main effects of *Extraversion*, $F(1, 33) = 1.82$, $p = .187$, ns, or *condition*, $F(1, 33) = 2.20$, $p = .148$, ns on the dependent variable of *Affective Trust*. However, again a significant interaction effect was found between *Extraversion* and *condition* on *Affective Trust*, $F(1, 33) = 5.56$, $p = .024$, $\eta^2 = .144$. Participants who scored low on *Extraversion* reported lower *Affective Trust* towards their fellow participant when using a joystick ($M = 2.29$, $SD = 0.95$) than when using a Frebble ($M = 4.25$, $SD = 2.22$; Fig. 7B).

## DISCUSSION

This study was performed to investigate whether mediated hand touching can have some of the same beneficial effects as reported for interpersonal touch. Therefore participants successively watched two emotional video clips and were led to believe that a fellow participant in an adjacent room simultaneously watched the same clips. The first video clip was intended to induce sadness while the second one was intended to cheer up the participants. While watching the second (funny) video clip the participants were asked to signal their fellow participant whenever they were amused, either via a mediated hand touch device or via a joystick. They received either haptic or visual feedback: whenever their fellow participant found the video amusing they received a mediated hand press (experimental condition) or a visual cue (a gray square on the screen turned blue—control condition). In reality there was no fellow participant present and a computer sent standardized feedback that had been scripted in advance.

It was expected that, compared to the control condition, communication with mediated hand touch feedback would (H1) enhance recovery from the sad mood, (H2) intensify the affective experience of the funny movie clips, and (H3) increase trust towards the fellow participant.

### Experimental manipulation

Analysis of the physiological measurements showed that the funny video clip indeed successfully elicited the desired emotion of amusement. While watching the funny video clip the participants' heart rate and skin conductance level increased significantly compared to both their baseline levels and their levels during the sad video clip. The selected sad movie only resulted in moderate emotional effect: heart rate and skin conductance level both decreased relative to their baseline levels but this difference was not significant. An explanation for the different extent of the physiological reactions to respectively the sad and funny video clips may be found in the dimensionality of emotions. Emotion has been conceptualized as a dimensional construct with valence and arousal as the main dimensions (*Mehrabian & Russell, 1974*). Valence refers to whether an experience is pleasurable (pleasant vs. unpleasant), and arousal refers to autonomic arousal associated with the experience (relaxed vs. aroused). In this view sadness can be described as a negative experience with a neutral level of associated arousal, while amusement can be described as a pleasant experience with a high level of associated arousal. The stronger physiological reactions to the funny video clip are probably due to a higher level of arousal, whereas the neutral arousal level induced by the sad video clip probably resulted in weaker physiological responses (*Fernández et al., 2012*; *Gunes & Pantic, 2010*). Here we should note that results in the literature on skin conductance responses to watching sad movies are also not unequivocal (both increases, decreases and the absence of skin conductance responses have been reported: *Kreibig, 2010*). Therefore it is also important to take the self-report questionnaires into account. The participants reported moderately high sadness scores on the sad video clip and moderately high amusement scores on the funny video clip. It can therefore be concluded that the induction of amusement was successful both

on a subjective and on an objective level, whereas the induction of sadness was confirmed on a subjective level. Although this is not the most extreme manipulation thinkable, we consider the effects sufficient for the current experiment.

## Recovery from sad mood

Previous studies indicate that interpersonal touch leads to a faster recovery from a sad mood or anxious events (*Debrot et al., 2013*; *Whitcher & Fisher, 1979*). However, this effect has not been widely examined for mediated touch (*Haans, de Bruijn & IJsselsteijn, 2014*). Only one study (*Cabibihan, Zheng & Cher, 2012*) reported a faster recovery during a resting period after watching a sad movie with both human and mediated touch. The present study tried to replicate this effect for mediated hand touching and to extend our understanding of this effect. But contrary to prior expectations, communication via a mediated touch device did not lead to enhanced recovery from a negative mood induced by watching a sad movie clip. Participants in both conditions showed a clear increase in heart rate and skin conductance level when watching the funny video clip (as expected), but the extent of the increase was not significantly different between the visual and mediated hand touching feedback conditions.

A possible reason that there was no difference in recovery between the two experimental conditions in the present study could be due to the fact that the mediated touch used here differed in one important aspect from the one used by Cabibihan and colleagues (*2012*). In Cabibihan's study the mediated touch device did not only provide haptic stimulation (like the Frebble did in the present study), but also provided warmth. A lack of warmth could cause the unsuccessful replication of their results in the present study. Generally, the effect of pleasant touch (strokes) is larger for stimuli at skin temperature (*Lucas et al., 2014*).

As mentioned before, the communication of amusement from the participant to the imaginary fellow participant was similar in both conditions (pressing the button on the joystick versus squeezing the Frebble). The main difference between both groups was the feedback mode, which was either visual or haptic. It is known that simple signals may be sufficient to mediate affective communication (*Janssen, IJsselsteijn & Westerink, 2014*) and that sharing feelings per se may already account for a certain amount of recovery. This effect could not be measured in this study because both groups communicated their feelings. Hence, the fact that participants in both conditions were able to communicate their feelings could account for the main effect independent of the actual nature of the mediated feedback (visual or haptic).

## Affective experience

This study also investigated whether the affective experience itself can be influenced by the use of a mediated hand touching device. Subjective measurements in the form of self-report questionnaires and objective measurements like number and duration of presses and facial expressions served to assess the affective experience. None of these measures revealed significant differences between the control and experimental condition. However, all objective measurements showed a marginal trend towards a more intense affective experience with the mediated touch device. Participants in the mediated touch
condition smiled for longer periods and pressed the device more often and longer than participants in the control condition. Note that although these measures are taken to characterize a participant's affective response to the video clip, they could also be influenced by the novelty of the interface. Despite these trends, it remains unclear why the use of the mediated touch device did not lead to a more intense affective experience.

The absence of a more intense affective experience may imply either that this effect cannot be induced by mediated touch at all or that the used form of mediated touch in this study was inappropriate. It is therefore important to investigate other aspects of mediated touch (like warmth) to get more insight about the possible effects of mediated touch on affective experience. Another element of mediated touch which is known to influence affective communication, but which was not tested in this study, is visual stimulation. It has been found that adding a mutually shared haptic sensation to a video conferencing system can significantly enhance the experience of social telepresence and mediated touch, provided that the mediated social touch is mutual but not visually duplicated (i.e., one should be able to see the partner and his movements to grasp the intention of the touch, but not one's own body part that is being touched, since this disturbs the illusion: *Nakanishi, Tanaka & Wada, 2014*).

The absence of a difference between the control and experimental conditions may also be due to the way feedback was provided during this study. Previous studies show that the quality of a conversation may depend on the richness of its content (*Haans & IJsselsteijn, 2006*). In this study the participants used a very simple, non-rich tactile code to communicate their feelings to their fellow participant (one press = feeling amused) in both conditions. Although the haptic device was able to deliver richer touch cues, we deliberately reduced the richness of its code to the level of richness of the visual cue (to make both feedback modes more comparable). Also, the feedback of the fellow participant was clear and unambiguous (color change of a square or haptic stimulation = feeling amused). The only difference between both conditions was the channel used for the feedback: haptic feedback in the experimental condition versus visual feedback in the control condition. It could be that the absence of a difference between both conditions means that the most important requirement for a shared affective experience is the ability to communicate feelings to another person, irrespective of the communication mode (as also suggested by *Spapé et al., 2015*). If so, haptic feedback has no benefit over visual feedback for affective communication. This hypothesis could be verified by including a condition in which the participants cannot communicate their feelings. Finding (1) a difference between conditions in which feelings are either shared or not shared, and finding (2) no difference between conditions in which feelings are shared through different communication channels, would confirm the idea that the mere ability to share feelings with another person is sufficient to mediate an affective experience (*Spapé et al., 2015*). However, this does not exclude that increased haptic communication effects may occur when the haptic messages become richer.

In addition, the mediated haptic feedback provided through a Frebble device (which resembles more a mechanical contact than the touch of a real human hand) may require

cognitive effort on the part of the receiver to be understood and may therefore have lost its unique affective quality. Although laboratory studies indicate that emotional information can to some degree be transmitted between two persons via (even very simple) mediated haptics (*Bailenson et al., 2007*; *Haans & IJsselsteijn, 2009*; *Rantala et al., 2013*; *Smith & MacLean, 2007*), and that real and simulated mediated handshakes are similar to some degree (*Giannopoulos et al., 2011*), it is still unknown to what extent mediated haptic experiences can actually approach direct interpersonal haptic interactions (*Gallace & Spence, 2010*).

Participants in both conditions reported that the constant feedback of their fellow participant led them to press their device more often than they would have done without feedback. They felt committed to reciprocate the presses of their communication partner. This may indicate a high social conformity in both conditions. Social conformity means that people change their behavior and confirm to the expectations of the other person (*Aarts & Dijksterhuis, 2003*; *Aronson, Wilson & Akert, 2010*; *Kiesler & Kiesler, 1969*). If so, a ceiling effect may have occurred caused by the set-up with pre-scripted communication, which makes it difficult to find differences in press behavior between the experimental and control condition.

Although there was no difference in affective experience the arousal level of the participants in the experimental condition (the Frebble users) was significantly higher after watching both movie clips compared to the arousal level before the experiment. In contrast, the arousal level of the joystick users decreased. A high arousal level can indicate two emotional states: stress or excitement (*Russell, Weiss & Mendelson, 1989*). In combination with positive feelings measured after the funny movie clips, it most likely indicates excitement here. This result suggests that mediated touch does not influence the valence of the affective experience directly, but only indirectly through increasing arousal levels. It may be that haptic communication in itself is arousing (especially with an unfamiliar partner), or that the use of a new haptic gadget is interesting and exciting for the participants and thus increases the self-reported arousal level.

## Trust

Finally the Hypothesis H3 that mediated touch increases trust towards another person more than abstract visual feedback was tested by playing the trust game. This game is designed to measure trust towards another person (*Berg, Dickhaut & McCabe, 1995*; *Kreps, 1990*; *Lazzarini et al., 2004*). When used in combination with self-report questionnaires it can provide an accurate and reliable picture of trust (*Glaeser et al., 2000*). However, the average amount of money set in, as well as self-report questions concerning the other person, showed no indications that mediated touch enhanced feelings of trust relative to visual feedback. Because it is known that social touch generally tends to increase trust (*Bailenson et al., 2007*), the question arises whether the lack of an effect indicates that some essential elements of mediated touch were missing in this study, with the result that mediated social touch was not able to affect trust. Again, it is also possible that the ability to share feelings is by itself already sufficient to establish a certain level of trust

(independent of the mode of communication) and that the difference between visual or mediated feedback therefore does not add any extra value. Another option is again the lack of warming actuators. Using warmth as an element of mediated touch is not only important for building the feeling of comfort, it is also essential for building trust (*Cabibihan et al., 2010*; *Haans & IJsselsteijn, 2006*).

## Limitations

Although the present study provides new insights on the effects of mediated touch some limitations should be mentioned. The experiment took place in a laboratory setting with only binary (visual or tactile) feedback. The fully scripted protocol did not resemble a natural conversation between two persons, and made it complicated to test acquainted dyads. The advantage of this controlled set-up is that the results are analyzable in a standardized way. However, a disadvantage is that the content of the conversation is strictly limited by the experimental paradigm and does not resemble real life content (*Haans, de Bruijn & IJsselsteijn, 2014*). Although squeezing each other's hand over distance while watching a funny movie may currently not be a common practice, there could be (subtle) effects of touch which might make this new technique interesting for certain applications. Note that techniques like texting and apping were initially also considered highly unnatural while they have become common practice in many situations nowadays. A second limitation of this study is that the experiment was not conducted with two persons who are related with each other or who have even met before. Previous studies have shown that the effects of mediated touch strongly depend on the familiarity of the persons and their relationships (*Bickmore et al., 2010*). In general, people consider mediated touch only appropriate as a means of communication between partners in close personal relationships (e.g., *Rantala et al., 2013*), and even mediated touch communication between strangers can cause discomfort (*Smith & MacLean, 2007*). Although feelings of amusement can effectively be communicated between unacquainted dyads by touching hands (*Hertenstein et al., 2006*) and although (mediated) social touch can positively affect human feelings and behavior even between strangers (e.g., the Midas touch; *Haans & IJsselsteijn, 2009*), touch may not be the right modality to share feelings of sadness or amusement between strangers. It is also uncertain whether the illusion of a fellow participant really worked and whether all participants believed that there was a fellow participant in the other room. This was intentionally not checked afterwards to avoid raising suspicion which might be communicated to future participants. A third limitation was the fact that participants reported that a Frebble squeeze resembled more a mechanical contact than a real human hand. In addition, squeezes may not have been the most appropriate form of interaction, since they are typically associated with unpleasant and aroused emotional intentions, while finger touch (stroking, tickling) is more often associated with pleasant and relaxed emotional intentions (*Rantala et al., 2013*). However, given the fact that simple mechanical pokes can effectively convey affective messages and elicit a Midas Touch effect (*Haans & IJsselsteijn, 2009*; *Spapé et al., 2015*), it has been suggested that it is not so much

the type of touch but more its meaning that determines the receiver's response in a certain context (*Haans & IJsselsteijn, 2009*; *Hoggan et al., 2012*).

## Future research

The mediated touch device used in this study was a prototype and should be improved before being used in further studies. Other studies indicate electromechanical actuators alone may not simulate accurately the feeling of being touched by a human hand (*Haans, de Bruijn & IJsselsteijn, 2014*). Therefore it would be beneficial to add warmth to the device as well as to a more humanlike surface of the device (softer and smoother). *Bickmore et al. (2010)* also state that mediated touch should resemble more a stroke than a pressing contact. The form of the device (which evoked the feeling of another hand enclosing the own one), as well as the fact that the device was held in the hand, was evaluated during the debriefing as positive by the participants and should not be changed. This impression corresponds to studies which confirm that touching hands is associated with loving, friendly and pleasant feelings (*Bickmore et al., 2010*).

Second, it may be interesting to conduct research in a more natural setting. First of all this involves the use of participants who know each other. On the one hand this may lead to more realistic communication and on the other hand it is known that this can improve the effects of mediated touch (*Tolmie & Boyle, 2000*). In a future laboratory study it is recommendable to use the mediated touch device in an open and more realistic conversation rather than a strict and limited conversation with only binary feedback as used in the present study. For a field study the mediated touch device could be integrated into the daily lives of dyads who know each other well and regularly use mediated communication. It is also known that long term effects of emotions are different from short living emotions, like film induced emotions (*Haans & IJsselsteijn, 2006*). Therefore, it could be useful to conduct a longitudinal study wherein mediated touch communication is observed over a longer period and can be compared to communication without a mediated touch device. Including these aspects will lead to a more natural context and thereby increase the ecological validity of the study.

Third, the present study used touch in isolation from vision and sound. It is known that visual and auditory cues are the major factors that influence the outcome of virtual communication (*Hammick & Lee, 2014*) and that the provision of facial expressions and speech enhances arousal and valence during a conversation (*Bickmore et al., 2010*). Although previous studies have shown that specific emotions can be encoded and decoded using touch, the effectiveness of touch in real life may also be in its complementary role to visual and auditory information. It would therefore be interesting to investigate how mediated touched moderates communication through other channels.

Finally, exploratory analyses of the present results revealed some interesting findings on the effects of *Extraversion* and *Touch Receptivity*. Using mediated touch via a Frebble device enabled participants scoring low on *Extraversion* or on *Touch Receptivity* to achieve the same level of trust towards their communication partner as participants scoring high on these factors, while they only achieved much lower levels when using a joystick. This

suggests that people who are not extroverts or who do not like to touch other persons felt better understood by their fellow participants when using mediated touch. These findings agree with the observation that introverted people communicate their 'real me' on the internet (i.e., in the absence of physical interaction) more easily than in real (physical) social interactions (*Amichai-Hamburger, Wainapel & Fox, 2002*), are more inclined to form online relationships (*McKenna & Bargh, 1999*), and feel more successful in online than in face-to-face interactions (*Shalom et al., 2015*). Since the present results are only explorative, future research should try to gain more insight on the effects of *Extraversion* and *Touch Receptivity* on mediated touch communication.

## CONCLUSION

This study investigated whether mediated hand touching has the ability to (1) enhance recovery from sadness (i.e., the return to a more positive emotional state) after watching a sad movie, (2) enhance a positive experience (watching a funny movie), and (3) increase trust towards the communication partner. The analysis of objective and subjective measurements showed no significant benefits of mediated touch communication (two-way tactile communication) above a control condition (tactile response and visual feedback). The present results do not allow us to conclude whether mediated touch is not able to do so in general or whether the used mediated touch device with only haptic stimulation is not sufficient and should be adapted. In both experimental conditions participants used a haptic channel to express their emotions. Only the feedback modality differed and was either visual or haptic. Giving and receiving mediated feedback can play a different role and giving input may already be sufficient to achieve the desired effects. This study therefore only allows conclusions about the reception of mediated touch. All-in-all, we were not able to replicate earlier favorable effects of mediated touch. This may indicate that these favorable effects may only occur under specific conditions. Future research should investigate the influence of personality, type of relationship with the other person, the effects of giving mediated versus non-mediated feedback, the role of intention, and the added value of respectively warming actuators and the visualization of the conversation partner to mediated touch.

The limitations of this study were that the participant was unfamiliar with the fellow participant and that the device was used in a limited and unnatural conversation. Besides adapting the mediated touch device, a suggestion for further research is to use a realistic conversation between dyads who are acquainted.

### Funding

The authors received no funding for this work.

### Competing Interests

The authors declare there are no competing interests. Stefanie M. Erk, Alexander Toet and Jan B.F. Van Erp are employees of TNO.

## Author Contributions

- Stefanie M. Erk conceived and designed the experiments, performed the experiments, analyzed the data, contributed reagents/materials/analysis tools, wrote the paper, prepared figures and/or tables, reviewed drafts of the paper.
- Alexander Toet conceived and designed the experiments, contributed reagents/materials/analysis tools, wrote the paper, prepared figures and/or tables, reviewed drafts of the paper.
- Jan B.F. Van Erp conceived and designed the experiments, wrote the paper, reviewed drafts of the paper.

## Human Ethics

The following information was supplied relating to ethical approvals (i.e., approving body and any reference numbers):

TNO Internal Review Board.

## Supplemental Information

Supplemental information for this article can be found online at http://dx.doi.org/10.7717/peerj.1297#supplemental-information.

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
