# Peer review of "Effects of mediated social touch on affective experiences and trust"

_PeerJ, doi:10.7717/peerj.1297_

## Round 0.1 · original submission · Major Revisions

I think that both reviewers are trying to highlight issues in the study that might potentially increase its validity and impact. I would like to invite you to carefully read them and try to incorporate as many suggestions as possible in any revised version of the manuscript.

·

Basic reporting

The article is by and large well written. The aims, method and results are described and reported with sufficient clarity and completeness. However, the exploratory analyses should be described with more clarity (Line 586 and onwards). What was the actual test that is described on line 590? Several ANCOVAs were conducted per dependent variable with age, gender etcetera as covariates, but were experimental condition or time of measurement included as factors? Where all covariates added in one single ANCOVA? Given how these exploratory analyses are reported, it would be difficult to reproduce them. Also, what is the difference between these analyses and those described in the section starting with Line 594, other than that the covariate was median split into a dichotomy? And why was that median split needed?

Second, the authors are advised to check carefully their use of references. I will give three examples. First, on line 703 the authors use Haans and IJsselsteijn (2006) as a reference to the claim that media richness is important for interaction quality. In my reading of that article, they actually warn the reader that that is not necessarily always true. Second, on line 107 the authors refer to Hertenstein and colleagues (2009) when stating that touching hands is a natural mode to communicate feelings of amusement. However, in my remembering of that article it only tells us that people can encode (and decode) amusement in a touch, not that it is the most natural way to do it by means of touch. Third, on line 821 the authors state that “It is known that visual and auditory cues are the major factors that influence the outcome of virtual communication” and use amongst others Haans, de Bruin and IJsselsteijn (2014) as a reference even though no such claim is made by these authors. Some additional minor issues are provided in the general comments.

Experimental design

The authors clearly invested time and effort in designing the experiment, and have provided a thorough argumentation behind most of the design decisions. The limitations in the interpretation and generalizibility of the results that stem from these decisions are by and large sufficiently discussed. My main concern with the experimental design is that it remains speculative whether the context of watching and communicating about videos represents a situation in which touch naturalistically is used in the Dutch population. Similarly, although I am not familiar with trust game literature, I wonder whether the context of the experiment and the one shot (one round) trust game sufficiently creates a situation in which trust becomes important. Both these aspects, although briefly mentioned in the discussion, should in my opinion be more strongly emphasized. Some additional minor issues are provided in the general comments.

Validity of the findings

The experiment did not reveal statistically significant effects of mediated social touch. However since the experiment was well designed using, for the most part, established measurement instruments, the results can be considered valid albeit of course completely within the specifics of this experimental design. However since the authors provide an elaborate discussion of the possible limitations of their experimental design, the experiments offers interesting materials for discussing the possible boundary conditions within which mediated touch may be effective. Admittedly, I did not try to replicate the authors’ analyses on the basis of the provided data, but some additional minor issues are provided in the general comments.

Additional comments

The authors present an experiment on the possible beneficial effects of (receiving) mediated touch on recovery from a sad event, affective reactions and interpersonal trust; effects that have been associated, albeit with varying degrees of empirical evidence, to real unmediated social touch. Their results do not support the anticipated effects of mediated touch. Nevertheless, such empirical findings are important to the field as empirical validation studies remain rare, and are often inconclusive or in disagreement. This disagreement shows that boundary conditions are yet largely unknown. As a result it is important that such studies are published, even when not yielding statistically significant results. I reviewed an earlier version of this manuscript when it was submitted to a different journal, and I am glad to see that the authors substantially improved their manuscript and have resubmitted it elsewhere. I recommend accepting this manuscript for publication in PeerJ, but advice the authors to first revise it according to issues raised in this review.

Minor issues:

Abstract and elsewhere: I do not see the term galvanic skin conductance being used often, if at all, and I wonder whether it is correct. Please check, or use the more common skin conductance instead (and when appropriate skin conductance level).

Line 94. The authors refer to studies on the Midas touch as evidence for the hypothesized role of touching in building trust. To my knowledge, however, it has not been demonstrated that trust drives the Midas touch phenomenon (in fact various non-trust related explanations have been proposed). Therefore it may not be correct to refer to the Midas touch effect as evidence for touch affecting trust.

Line 107. It is stated that (3) touching hands is a natural mode to communicate feelings of amusement. From Hertenstein’s study we may conclude that people can communicate amusement via touch, but not so much that it would be a natural way of communicating amusement with a stranger (the ostensible partner in the experiment must have been a stranger). I think this is particularly so in the Netherlands. Similar arguments can be made for (1) people feel a fundamental need to share their feelings after a sad event. The literature that the authors discuss (Line 72 and onwards) mainly reports on communication between friends. Although the discussion mentions this possible limitation, it should be emphasized a bit more, and a discussion of the referenced literature with respect to whether friends or strangers were used seems relevant here.

Line 118. The manuscript still contains some redundancy in the method section that should be avoided. To give one example: it is mentioned multiple times that a grey and blue square is used in the no touch conditions (e.g., Lines 151, 174, 216, and 319).

Line 183. Please remove the “; see”

Line 209. “Frebble feedback was thresholded at a level of 600”. It is not sufficiently clear what that means. I expect 600 is the pressure of the lever which applies force to the back of the hand. Please revise this section so that it is clear that feedback was scripted as a dichotomy with everything below 600 as no feedback and above as feedback with a force level of 800. Would this render the feedback more similar to the joystick condition because the visual feedback was dichotomous as well? Please explain in the text more clearly why it would make the two conditions more similar.

Line 253. Please replace “are associated with” with something like “have previously been associated with” or something similar that makes it clear that these are not yet established measures. Since these are not yet established measures it would be beneficial if the authors could report the correlation between number of button presses / duration and other measures used to quantify the intensity of the affective experience.

Line 450. The text states that the baseline was statistically significant from the funny video but this is not indicated in Figure 4b

Line 485. What was the reason for measuring SCL only during the first 90s of the funny movie? Please explain.

Line 480. Please include appropriate effect sizes with non-significant results.

Line 496. The sentence “there were no significant differences in …… in both conditions over the period between the lasts four minutes of the Champ and the 90…” is a bit vague. I would expect that the effect of mode of feedback on recovery is tested by the interaction. Please rephrase.

Line 599. “All effects were statistically significant at the .05 significance level. It is chosen to only report significant results of the factorial ANOVAs.” This sentence should be clarified and rephrased. Were other analyses beside factorial ANOVAs conducted? Or were all hypothesis tests performed with a confidence level of .05 but only the significant ones are reported? Since these exploratory analyses involve quite a large number of tests, did the authors consider using a correction to avoid capitalizing on change? If not, why not? This should be included in the manuscript.

Line 685. Please include that increased smiling and use of the communication tool (i.e., number and duration of presses) in the Frebble condition may also be a direct result of the interface (e.g., its novelty) rather than due to an increased affective response to the video.

Line 825, consider adding “or how mediated touched moderates communication through other channels”. The main point to make here is that although studies have indicated that specific emotions can be encoded and decoded using touch, the effectiveness of touch in real life may also be in its complementary role to visual and auditory information.

Line 827. I agree that findings related to Extraversion may make sense, but how should we explain the touch receptivity finding. Is the direction of the effect not against expectations? Also the authors should state once more the exploratory nature of these findings.

Line 857. In the list of possible boundary conditions that need to be investigated, please include the type of relationship with the other person, and the role of intention.

·

Basic reporting

The basic reporting is sufficient.

Experimental design

I had serious doubts about the ecological validity of the situation in which participants found themselves. Basically, they were asked to watch a movie, and that the researchers were interested in investigating people’s emotions. First, the study did not seem very ecologically valid, as people do not readily engage in such interactions with the Frebble (the measurements could not confirm whether it was ecologically valid or not, later more on this). Second, I have great concerns about demand characteristics. In studying this, it would be better to give a cover story for the study (not the studying of emotional experiences), and then do a so-called funneled debriefing, to gage participants’ suspicions towards the purpose of the study. In addition, they asked for touch receptivity prior to their DVs. This likely cues the participant to pay attention to specific details of the study. All of these doubts are quite fundamental, and cannot be repaired with a revision of this paper.

Validity of the findings

See below.

Additional comments

Participant selection
It was unclear to me how they determined their sample size. With the recent discussions in psychological science, it is very clear that we need to plan better how we collect our participants. Specifically, I had expected an a priori power analysis (e.g., via G*Power). From reading this paper, it was also clear that their randomization had failed. Even with this small sample, their seemed to be a difference in extraversion across the conditions (despite marginal significance, but this is to be expected with this sample size). In addition, they reject a number of hypotheses, but this is not possible with this limited N. Then studying the interaction simply leaves for random effects that are likely simply due to chance.

Analyses
The repeated measures analysis is possible, but not the best analysis. In case of measurements like heart rate, one should use a linear mixed model, because the variance across measurements is likely different, and a LMM is a bit less conservative. In addition, it leaves for a bit greater predictive power in making the inferences that the authors would want (note: not all analyses were entirely clear to me, like the time of measurement analysis on heart rate in line 480).

Theoretical Introduction
At present, the theoretical introduction was a bit too scattered. They discuss the sharing of emotions, empathy, affective touch, and so forth. The introduction really needs some cleaning up, and can likely simply focus on how people deal with homeostasis (and the regulation of stress, see e.g., the work by Lane Beckes and/or Jim Coan).

I did like the general idea of the study a lot, and there is something there in terms of ideas. They were also very careful in terms of designing the study. However, it is clear that the design should be better thought out, and that the theoretical introduction needs to be cleaned up. They could consider registering their study in advance as well on the Open Science Framework prior to running a second study, which would add to the convincingness of the study. I am sorry I cannot provide better news, but highly encourage the authors to continue this interesting program of research.

---

## Round 0.2 · accepted · Accept

This is an interesting case of cultural differences (and perhaps emotional hijacking). I agree with Hans that not giving a cover story makes the experimental subjects more prone to be sensitive to the experimental condition. But isn't this exactly what the Authors want to prove 'in the wild'? If we ever decide we want to use these devices in our everyday life, it's because we crave for those feelings/emotions. So, I don't see this aspect of the design as a fatal flaw. Indeed, I'd be worried that if a version of this study with the cover story that Hans wants shows no experimental effect, its relevance to real life would be limited.

On sample size I disagree with the Authors (on two accounts: one is the it's the significance of the outcome measure that determines how even very mild effects are important - in early stroke therapy any tiny decrease in mortality rate or tiny increase in recovery of function is huge, no matter how big is the sample size; the other one is that small sample sizes always have the winner's curse problem, as Ioannidis obsessively reminds us). Yet, the sample size of this study isn't obscenely small. It aligns to standard practices (which aren't good, as we all know).
I see no reason to prevent publication of this study.

·

Basic reporting

Good, but still I feel that the authors use of references maybe improved. See minor issues below.

Experimental design

All previous expressed concerns and limitations have been addressed in the revision

Validity of the findings

All previous expressed concerns and limitations have been addressed in the revision

Additional comments

I thank the authors for their thorough revision of the original manuscript and by and large the main concerns I had with the original version have been addressed. Only a few minor issues remain, and I recommend accepting the manuscript for publication in PeerJ with only minor revisions.

Minor issues:

Line 139. Remove the space before the , in “on a joystick , and they received”

Line 641. In (H1) the last ) is in italic

Line 811. Both Haans and IJsselsteijn (2009) and Spapé et al (2015) have used vibrotactile stimulation, which, at least to me, is something different than a “mechanical poke”. Please revise.

Line 835. The authors have revised many of the imprecise uses of references in the present version. Still some references seem out of place. For example, here the authors refer to Haans and IJsselsteijn (2006) when claiming that short lived emotions are different from long lived emotions. However, I cannot remember such a discussion in that paper, and it certainly was not a central argument in that manuscript.

·

Basic reporting

I will paste all my commentary from the word document here. This includes comments to the editor.

I regret having to write this review. In my initial review, I think I as relatively mild, but I think the paper has major methodological shortcomings. Because the editor was inattentive to the reviews, I will have to point the flaws out point by point. I think it is particularly unnecessary when the most major points have been attended to. So, my apologies to the authors, and I am surely willing to consult with them for a future study.

Editorial Decision:
Editor
I think that both reviewers are trying to highlight issues in the study that might potentially increase its validity and impact. I would like to invite you to carefully read them and try to incorporate as many suggestions as possible in any revised version of the manuscript.
-- Fulvio D'Acquisto

Response IJzerman:
In this particular case, there were two opposing reviews. That can happen. However, I don’t think that this is an acceptable response from an editor. The editor should weigh both sources of input and make a recommendation, in particular because the views were so divergent. Below a response to my request for clarification on this decision. As a reviewer, I am quite disappointed about the process. I will start with the most pressing concern.

Clarification Editor:
Editor
In agreement with this view, the other reviewer commented: "it is important that such studies are published, even when not yielding statistically significant results."
I fully agree and support this statement as many other colleagues and lay public members do in current times (see some links below)
Opinion: Publish Negative Results | The Scientist Magazine®
Why science needs to publish negative results
On the importance of being negative | Science | The Guardian

Response IJzerman:
I do not even understand why the editor cites this. I did not cite the significance level of the study at any time for reason for rejection. My criticisms were focused on the methodological flaws of this study, not AT ALL on significance level. In fact, the editor can look up a recent null result publication from yours truly, IJzerman, Regenberg, Saddlemyer, & Koole, 2015). It surely is a straw man. This means that either 1) the editor did not read my review, or 2) wants all studies – irrespective of quality – to be accepted. I don’t know what’s worse.

Editor
I have listed below few points behind my decision:
1- First and fore most, we had two decisions on the manuscript: One 'reject' and one 'minor revision'. Looking at both comments it was my decision to give the authors a chance and hence propose 'major revision' as a true 'mean' - middle point- compromise of the reviewer's opinions;

Response IJzerman:
A mean would imply that per definition the responses are equal. I am not negating this possibility, and Haans’ response may be better than mine, but this needs to be elaborated upon. I am fine with being wrong, but please indicate what the flaws are in my review. This is also helpful for the authors.

Editor:
2- the other reviewer stated "The authors clearly invested time and effort in designing the experiment, and have provided a thorough argumentation behind most of the design decisions. The limitations in the interpretation and generalizibility of the results that stem from these decisions are by and large sufficiently discussed."
I agree with these comments and I think that the moment an authors recognised the potential weaknesses of their approach, they simultaneously take responsibility of the potential limited impact of their study.

Response IJzerman:
This implies that any study, no matter how big the shortcoming (if I were to send in a study for which I would have fabricated my data for example), should be published (see Wigboldus & Dotsch, 2015). My question to the editor is if this is indeed true. If so, then I think we have a major problem.

Editor:
While I surely appreciated the pointed you have raised, such as "ecological validity of the situation in which participants found themselves. Basically, they were asked to watch a movie", I also think that this might be a subjective point of view that has not been supported by any referenced paper.

Response IJzerman:
When one provides a review, and there are such severe shortcomings of the study, it would be bad form to point out all the major mistakes in detail. And this concerns basic research design. In order to learn about research design, there are many directions one could take (e.g., a chapter by Brewer, 2000; but I would even more so recommend work by Brunswik, e.g., 1943, 1956). Again, I am surely willing to help out with designing a study that does so. A good example of an ecologically valid study is for example Coan et al.’s (2006) handholding study.

Editor:
Same applies for your other comment "In addition, they asked for touch receptivity prior to their DVs. This likely cues the participant to pay attention to specific details of the study." There is no proof to me that this is the case (in fact you carefully used the word 'likely') and ...if this was the case.... participants could have been asked this question at the end of the test thus still providing another layer of observation to the overall study.

Response IJzerman:
This is also very basic research design, and it goes back to the activation of semantic networks (e.g., Schachter & Tulving, 1994). See also Schneider’s dissertation (2013) for a recent application of this. Asking for the manipulation check prior to the most important dependent variable is a basic mistake in research design. If this were the only mistake, I could live with it, as long as there is mention of this shortcoming in the discussion, and that this flaw may have affected the results.

Editor:
Last but not least ... you rightly call all these 'doubts' (rather than flaws or mistakes) and I fully appreciated this since the most important aspect of peer review is not to judge the validity of the study (as most of the current journals do) but rather to check the methodological soundness and to highlight other aspects of the investigation that might enhance/clarify/improve the impact of the study.

Response IJzerman:
Everything in science is probabilistic, and, again, it would be bad form to respond the way I do now. There are severe and major shortcomings in this article, and it should not be published.

Editor:
-"Participant selection-It was unclear to me how they determined their sample size."
- "Analyses -The repeated measures analysis is possible, but not the best analysis."
- "Theoretical Introduction- At present, the theoretical introduction was a bit too scattered. The introduction really needs some cleaning up, and can likely simply focus on how people deal with homeostasis"

Response IJzerman:
As you will see below, none of these were addressed satisfactorily.

Editor:
Are all points that could be addressed in a revised version of the manuscript without repeating the study from the very beginning.
Response IJzerman:
This really depends on how big the flaws are. If you had asked for a new study, I would agree and I would be happy to help.

Reviewer 2 (Hans IJzerman)
Basic reporting
The basic reporting is sufficient.
Experimental design
I had serious doubts about the ecological validity of the situation in which participants found themselves. Basically, they were asked to watch a movie, and that the researchers were interested in investigating people’s emotions. First, the study did not seem very ecologically valid, as people do not readily engage in such interactions with the Frebble (the measurements could not confirm whether it was ecologically valid or not, later more on this). Second, I have great concerns about demand characteristics. In studying this, it would be better to give a cover story for the study (not the studying of emotional experiences), and then do a so-called funneled debriefing, to gage participants’ suspicions towards the purpose of the study. In addition, they asked for touch receptivity prior to their DVs. This likely cues the participant to pay attention to specific details of the study. All of these doubts are quite fundamental, and cannot be repaired with a revision of this paper.

AUTHORS’ RESPONSE:
“the study did not seem very ecologically valid, as people do not readily engage in such interactions with the Frebble” => Since mediated touch technology in general and the Frebble in particular are not readily used or even available at this time we simply do not know (yet) whether people are inclined to readily engage in such interactions. Time will inform us in which situations mediated touch may become an accepted practice. Sharing emotions during media events may certainly be a candidate.
Response IJzerman: This is not a good explanation. When one wants to create a situation that is ecologically valid, then the best thing to do is to mirror an already existing paradigm that mirrors e.g. the holding of hands (see Coan et al., 2006). It is true that these situations may become a reality in some distant future, but does that mirror social reality at this point? Surely not. Thus, the best approach then is to mirror an existing hand-holding type of study, or first do a number of pilots to see whether this actually works (and, yes, the other parts of the design need to be addressed then too).
We used a ‘cover story’ that complied with the instructions given to the participants, e.g. with respect to signaling their emotion and the meaning of the feedback cues signaling the other participant’s experience. Related studies in this field have used similar protocols (e.g., Janssen, IJsselsteijn & Westerink, 2014). We think that cover stories that are not related to the instructions would have resulted in an apparent conflict. Suggestions for cover stories that satisfy both criteria and are in line with ethical guidance are very welcome.

Response IJzerman: Using a cover story is a very basic part of experimental design. The reason for doing so is to avoid demand characteristics. The classical example in this particular case is of course the Hawthorne effect. I presume that the authors are aware of this effect. In order to avoid observer effects – or demand characteristics – a cover story is crucial. There are many such examples if one wants to do a study like this, and one of the classic ones is a study by Strack et al. (1988). I am more than happy to consult with the authors for a future study.
Validity of the findings
See below.

Comments for the author
Participant selection
It was unclear to me how they determined their sample size. With the recent discussions in psychological science, it is very clear that we need to plan better how we collect our participants. Specifically, I had expected an a priori power analysis (e.g., via G*Power). From reading this paper, it was also clear that their randomization had failed. Even with this small sample, their seemed to be a difference in extraversion across the conditions (despite marginal significance, but this is to be expected with this sample size). In addition, they reject a number of hypotheses, but this is not possible with this limited N. Then studying the interaction simply leaves for random effects that are likely simply due to chance.

AUTHORS’ RESPONSE:
We are very well aware of the size of physiological effects and base our sample size upon this information and the balance between significance and relevance (see e.g. Brouwer et al., 2014a; Brouwer et al., 2014b; Hogervorst, Brouwer & van Erp, 2014). From an applied point of view, effects that do not reach significance with group sizes in the order of 16 or more participants are usually of limited applied relevance.
Response IJzerman: I don’t think this is a sufficient justification. There are now extent discussions on this particular issue, but this started with Cohen. This study needs a larger sample size to make any inference. In order to justify sample, please do a power analysis. For repeated measures, one can calculate the ICC. Alternatively, one can do a very conservative power analysis in G*Power (http://www.gpower.hhu.de/; Faul et al., 2007, 2009; see also Brandt et al., 2014).

The distribution of participants over the conditions remains random of course.
“they reject a number of hypotheses, but this is not possible with this limited N.” => We agree with this observation, and replaced “rejected” by “not confirmed”.

Response IJzerman: Distribution over cells is fine. But if the N is too limited, why even publish this? I highly recommend running a new study with this feedback.

Analyses
The repeated measures analysis is possible, but not the best analysis. In case of measurements like heart rate, one should use a linear mixed model, because the variance across measurements is likely different, and a LMM is a bit less conservative. In addition, it leaves for a bit greater predictive power in making the inferences that the authors would want (note: not all analyses were entirely clear to me, like the time of measurement analysis on heart rate in line 480).

AUTHORS’ RESPONSE:
We agree that preferred analysis is a matter of style or taste. We prefer the more conservative repeated measures.

Response IJzerman: In such types of measurements, variance differs across the different measurement points. There is thus a better analysis (which by the way also increases their power, and they could even refute my argument of low power above if they look at power post hoc).
The time of measurement is explained in the “Objective measures” section.
Theoretical Introduction
At present, the theoretical introduction was a bit too scattered. They discuss the sharing of emotions, empathy, affective touch, and so forth. The introduction really needs some cleaning up, and can likely simply focus on how people deal with homeostasis (and the regulation of stress, see e.g., the work by Lane Beckes and/or Jim Coan).

AUTHORS’ RESPONSE:
We believe the Introduction is well structured and we do not agree that it is scattered.
In the “Aim of this study” section we clearly delineate the scope of this study: to test whether mediated touch can
1. stimulate recovery from a sad experience,
2. enhance a positive experience and
3. increase trust.
Then we consecutively elaborate on each of these points explaining the need for personal touch and how mediated touch may
1. fill the current technological gap
2. stimulate recovery from a sad experience,
3. enhance a positive experience and
4. increase trust.
We end the Introduction by giving a brief overview of the study.
We acknowledge that homeostasis may also be a relevant framework but our approach to the current study is from a different angle. Our point of departure is mediated touch and the presumed or possible effects on sharing emotions and recovery from a sad event, and not so much on how people deal with homeostasis and the regulation of stress.

Response IJzerman: The introduction discusses touch and its importance. It discusses touch in relation to trust, recovery from negative experiences, enhancing of positive experiences, then continues into well-being. So far so good, and for a broad introduction, this is fine. However, one then expects that it gets more concrete. The authors go into mediated touch, and here we start expecting the explanation of some kind of mechanism. What is it of a touch that enhances for example recovery from negative experience?
The authors indicate that technology can enhance social presence – another concept. Is social presence related to trust? Or how is it related to the earlier concept of well-being? How does it enhance recovery from a negative emotional experience? From there on, another concept is introduced: “social interactivity”. What does this mean? And how does this relate to trusting someone? What is social interactivity? In the next sentence, “affective touch patterns” is introduced. What is this concept?
Then, the authors introduce the idea that mediated touch can indeed to some extent convey emotions. This is a very broad claim. Can mediated touch convey people’s sadness, anger, shame, guilt, et cetera? And, going from Wilson-Mendenhall’s et al.’s (2011) idea that emotions are constructed from social experience – how is it possible that for example social and non social fear are communicated? This is a claim that is too broad without specifying.
Yet, in the next sentence, the authors go from conveying emotions to “affective experiences”. Do they simply mean a positive versus negative affective state?
Next paragraph goes from touch to sharing emotions verbally. What is the relation between verbal sharing of emotions and touch? They surely are related, but it is a big jump from one to the other. In the same paragraph, they talk about receiver versus supporter – at this point, I am really lost. What will be the focus of the study? Will it be on sharing of sad feelings? Will it be of positive versus negative affect? Will it be about sadness versus other emotions?

I will jump to a next section on trust. They indicate that touch improves trust. If this would be a central concept, one may expect a mediation analysis (or, at least, a comparative test) in that for example touch increases trust, and therefore would also enhance recovery from a negative state. What I would also expect here is a broader discussion of trust, different measurements of trust (and, likely, I would actually expect a measurement of communal strength or communal sharing; communal strength or communal sharing allows participants in the relationship to focus on the others’ needs, and is typified by a merging of bodies, at least, that is what Alan Fiske proposes, and we also have found communal strength to moderate many of comparable effects).
In short – the introduction is confusing. It can be watered down by a lot, simply focusing on touching, trust, and emotional experiences, and what these different pathways are. If the authors would do so, they would also be able to design a better study, that focuses on 1) the right sample size, 2) the right pathways and measurements (they actually measured felt understanding, which is comparable, but not the same as trust).
I could continue, but I think the point is made.

I did like the general idea of the study a lot, and there is something there in terms of ideas. They were also very careful in terms of designing the study. However, it is clear that the design should be better thought out, and that the theoretical introduction needs to be cleaned up. They could consider registering their study in advance as well on the Open Science Framework prior to running a second study, which would add to the convincingness of the study. I am sorry I cannot provide better news, but highly encourage the authors to continue this interesting program of research.

AUTHORS’ RESPONSE:
Thank you for your advice about the Open Science Framework. This is a nice illustration of the multidisciplinary field of mediated social touch. The use of this platform is not common in our discipline (computer science) but we will certainly explore its value.

Response IJzerman: This is only a recommendation to improve upon the study for a next iteration, and it would be possible to let it be reviewed a priori (e.g., by going to CRSP), and avoid the problems given in the present version.

Experimental design

see above

Validity of the findings

see above

Additional comments

see above